# KLF10 integrates circadian timing and sugar signaling to coordinate hepatic metabolism

**Anthony A Ruberto**[1†], **Aline Gréchez-Cassiau**[1], **Sophie Guérin**[1], **Luc Martin**[1], **Johana S Revel**[2], **Mohamed Mehiri**[2], **Malayannan Subramaniam**[3], **Franck Delaunay**[1], **Michèle Teboul**[1*]

[1]Université Côte d'Azur, CNRS, Inserm, iBV, Nice, France; [2]Université Côte d'Azur, CNRS, Institut de Chimie de Nice, Nice, France; [3]Department of Biochemistry and Molecular Biology, Mayo Clinic, Rochester, United States

**\*For correspondence:**
Michele.Teboul@univ-cotedazur.
fr

**Present address:** [†]Center for Tropical and Emerging Global Diseases, University of Georgia, Athens, United States

**Competing interest:** The authors declare that no competing interests exist.

**Abstract** The mammalian circadian timing system and metabolism are highly interconnected, and disruption of this coupling is associated with negative health outcomes. Krüppel-like factors (KLFs) are transcription factors that govern metabolic homeostasis in various organs. Many KLFs show a circadian expression in the liver. Here, we show that the loss of the clock-controlled KLF10 in hepatocytes results in extensive reprogramming of the mouse liver circadian transcriptome, which in turn alters the temporal coordination of pathways associated with energy metabolism. We also show that glucose and fructose induce *Klf10,* which helps mitigate glucose intolerance and hepatic steatosis in mice challenged with a sugar beverage. Functional genomics further reveal that KLF10 target genes are primarily involved in central carbon metabolism. Together, these findings show that in the liver KLF10 integrates circadian timing and sugar metabolism-related signaling, and serves as a transcriptional brake that protects against the deleterious effects of increased sugar consumption.

## Introduction

The mammalian circadian timing system aligns most biological processes with the Earth's light/dark (LD) cycle to ensure optimal coordination of physiology and behavior over the course of the day (***Bass and Lazar, 2016***). At the organism level, circadian clocks are organized hierarchically with a central pacemaker located in the suprachiasmatic nuclei of the hypothalamus that receives the light input and in turn coordinates peripheral clocks via internal synchronizers including glucocorticoids and body temperature (***Saini et al., 2011***). The molecular mechanism underlying circadian clocks is an oscillatory gene network present in virtually all cells, which translates the external time information into optimally phased rhythms in chromatin remodeling, gene expression, post-translational modification, and metabolites production (***Dyar et al., 2018***; ***Mauvoisin and Gachon, 2020***; ***Rijo-Ferreira and Takahashi, 2019***; ***Robles et al., 2017***; ***Robles et al., 2014***; ***Weidemann et al., 2018***; ***Zhang et al., 2014***).

The extensive circadian regulation of processes linked to metabolic homeostasis, as well as how the cellular metabolism feeds back to the clock, has been highlighted in functional genomics studies (***Panda, 2016***; ***Peek et al., 2013***; ***Reinke and Asher, 2019***; ***Sinturel et al., 2020***). In mice, the disruption of clock genes leads to multiple metabolic abnormalities, including impaired insulin release, glucose intolerance, insulin resistance, hepatic steatosis, and obesity (***Bugge et al., 2012***; ***Jacobi et al., 2015***; ***Perelis et al., 2015***); conversely, diet-induced metabolic imbalance impairs circadian coordination in part by reprogramming the circadian transcriptome (***Eckel-Mahan et al., 2013***; ***Peek et al., 2013***). Experimental and epidemiological data support a similar reciprocal relationship

between circadian misalignment or disruption and metabolic disorders in humans (*Qian and Scheer, 2016*; *Stenvers et al., 2019*).

The liver is a major metabolic organ in which circadian gene expression is regulated directly by core clock and clock-controlled transcription factors, among which nuclear hormone receptors including REV-ERBs, PPARs, GR, FXR, and SHP play a prominent role (*Mukherji et al., 2019*). Krüppel-like factors (KLFs) form another family of transcription regulators, a majority of which are involved in the physiology of metabolic organs including the liver (*Hsieh et al., 2019*). Many of these KLFs are also clock-controlled genes in mouse liver (*Ceglia et al., 2018*; *Yoshitane et al., 2014*). This suggests that the KLFs act as additional factors for the circadian regulation of hepatic metabolism. For instance, KLF15 was shown to control the circadian regulation of nitrogen metabolism and bile acid production (*Han et al., 2015*; *Jeyaraj et al., 2012*). KLF10, first described as a TGF-β-induced early gene in human osteoblasts and pro-apoptotic factor in pancreatic cancer (*Subramaniam et al., 1995*; *Tachibana et al., 1997*), has been further identified as a clock-controlled and glucose-induced transcription factor that helps suppress hepatic gluconeogenesis (*Guillaumond et al., 2010*; *Hirota et al., 2002*). In *Drosophila*, the KLF10 ortholog *Cabut* has a similar role in sugar metabolism and circadian regulation (*Bartok et al., 2015*), which suggests that the function of this transcription factor is evolutionarily conserved.

A better understanding of the role of KLF10 in hepatic metabolism may have important translational implications in the context of the rising prevalence of nonalcoholic fatty liver disease (NAFLD) and fructose overconsumption (*Jensen et al., 2018*). In this study, using a hepatocyte-specific *Klf10* knockout mouse model, we show that KLF10 is required for the temporal coordination of various biological pathways associated with energy metabolism and has a protective role in shielding mice from adverse effects associated with sugar overload.

## Results

### Hepatocyte-specific deletion of KLF10 reprograms the liver circadian transcriptome

To understand the role of the circadian transcription factor KLF10 in hepatic metabolism, we generated a conditional *Klf10* knockout mouse model by crossing WT mice (*Weng et al., 2017*) with a mouse line expressing an inducible CreER$^{T2}$ recombinase under the control of the endogenous serum albumin promoter (*Schuler et al., 2004*; *Figure 1A*). Upon tamoxifen treatment of *Klf10$^{flox/flox}$*, *Alb$^{+/CreERT2}$* mice, we obtained a deletion of *Klf10* specifically in hepatocytes (hepKO). Both qPCR and immunoblot analyses confirmed the targeted reduction of the transcription factor in the liver of hepKO mice (*Figure 1B*, left, C). Residual expression was attributable to non-parenchymal cells as *Klf10* mRNA was not detectable in isolated primary hepatocytes (*Figure 1B*, right). As expected, we detected a robust circadian oscillation of *Klf10* mRNA in the liver of WT mice, with peak expression of the transcript occurring just before the LD transition, while in hepKO mice, rhythmic expression of *Klf10* was absent (*Figure 1D*). Body weight, food intake, and food-seeking behavior did not differ between genotypes, indicating that postnatal hepatocyte-specific deletion of KLF10 did not grossly affect the physiology and behavior of unchallenged adult *Klf10$^{Δhep}$* mice (*Figure 1—figure supplement 1A and B*).

We next performed RNA sequencing (RNA-seq) on livers collected every 3 hr across the 24 hr day from WT and hepKO mice entrained to a 12 hr:12 hr LD cycle (*Figure 2A*). Using MetaCycle (*Wu et al., 2016*), we found that ~12% of genes (1,664) displayed circadian expression in WT mice (*Figure 2B* and *Figure 2—source data 1*), a percentage that is consistent with previous reports (*Greenwell et al., 2019*; *Zhang et al., 2014*). In hepKO mice, we detected a similar percentage of genes with circadian expression (~12 %; 1633) (*Figure 2B* and *Figure 2—source data 1*). The phase distributions and relative amplitudes of rhythmically expressed genes in WT and hepKO mice were similar (*Figure 2C and D*). Strikingly, we observed rhythmicity of 1031 transcripts exclusively in hepKO mice, indicating that the deletion of KLF10, a known transcriptional repressor, results in de novo oscillation of otherwise non-rhythmically expressed transcripts (*Figure 2E and F*, *Figure 2—figure supplement 1B*, and *Figure 2—source data 1*). Approximately 37% (602) of genes displaying a rhythmic expression in both genotypes included various high-amplitude transcripts (*Figure 2E and F*, *Figure 2—figure supplement 1B*, and *Figure 2—source data 1*). We found the core clock genes within this subset (*Figure 2—figure supplement 1A and B*, *Figure 2—source data 1*), suggesting that hepatocyte

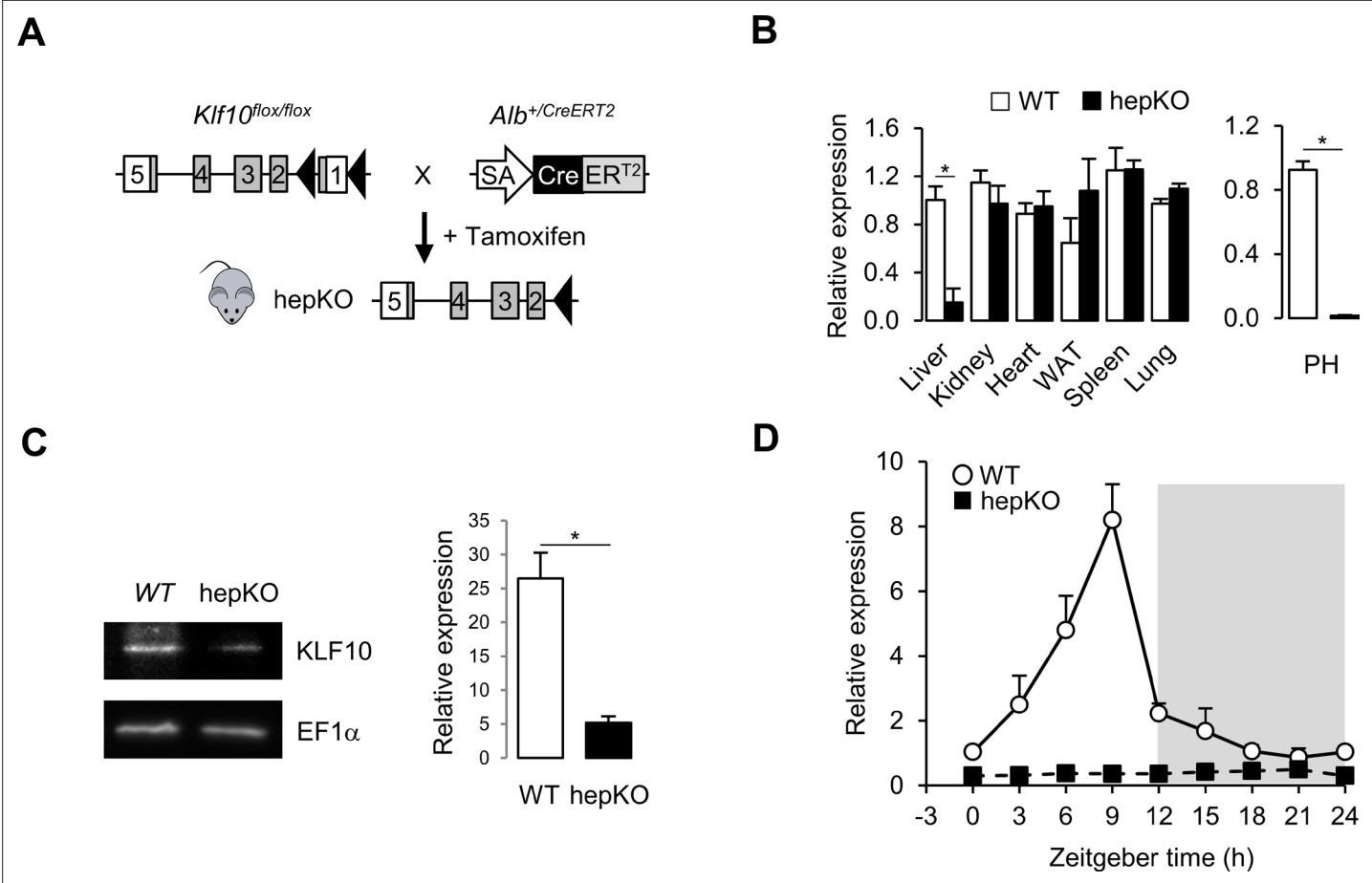

**Figure 1.** Genetic disruption of *Klf10* in mouse hepatocytes. (**A**) Schematic illustrating the strategy used for generating HepKO mice. (**B**) Relative gene expression of *Klf10* mRNA across various tissue in WT and HepKO mice as measured at ZT9 (left) and in WT and HepKO primary hepatocytes (PH) (right) (mean ± SEM, n = 3–5). (**C**) Immunoblot showing KLF10 protein abundance in liver extracts from WT and HepKO mice at ZT9 (left) and quantification of 3–4 independent experiments (right). EF1α was used as loading control. (**D**) 24 hr gene expression profiles of hepatic *Klf10* mRNA measured every 3 hr in WT and HepKO mice (mean ± SEM, n = 3 mice per time point). Statistics: nonparametric Wilcoxon test. *p<0.05.

The online version of this article includes the following figure supplement(s) for figure 1:

**Source data 1.** Values and statistical test results for *Figure 1B and D* and *Figure 1—figure supplement 1A*.

**Figure supplement 1.** Genetic disruption of *Klf10* in mouse hepatocytes.

KLF10 is not required for normal core clock function in the liver. A set of 1062 transcripts with a low amplitude in WT mice displayed a dampened rhythmicity in hepKO mice, resulting in a MetaCycle p-value above threshold (*Figure 2E and F*, *Figure 2—figure supplement 1B*, and *Figure 2—source data 1*). Together, this data indicates that the main effect of the hepatocyte KLF10 knockout is the derepression of the oscillatory behavior of a significant proportion of the liver transcriptome.

To better understand the changes in the circadian transcriptome associated with the deletion of *Klf10* in hepatocytes, we assessed the temporal coordination of biologically related transcripts in WT and hepKO mice. After performing Phase Set Enrichment Analysis (PSEA) (*Zhang et al., 2016*), we found a decrease in the total number of temporally enriched gene sets in hepKO mice (*Figure 2G*). Specifically, gene sets associated with processes such as oxidative phosphorylation, fatty acid metabolism, and mTORC1 signaling peaking around the LD transition in WT mice were absent in hepKO mice (*Figure 2G* and *Figure 2—source data 2*). Despite the global reduction in temporally coordinated transcripts, hepKO mice acquired de novo temporal enrichment of transcripts associated with glycolysis and cellular stress processes such as apoptosis, UV response, and inflammation (*Figure 2G* and *Figure 2—source data 2*).

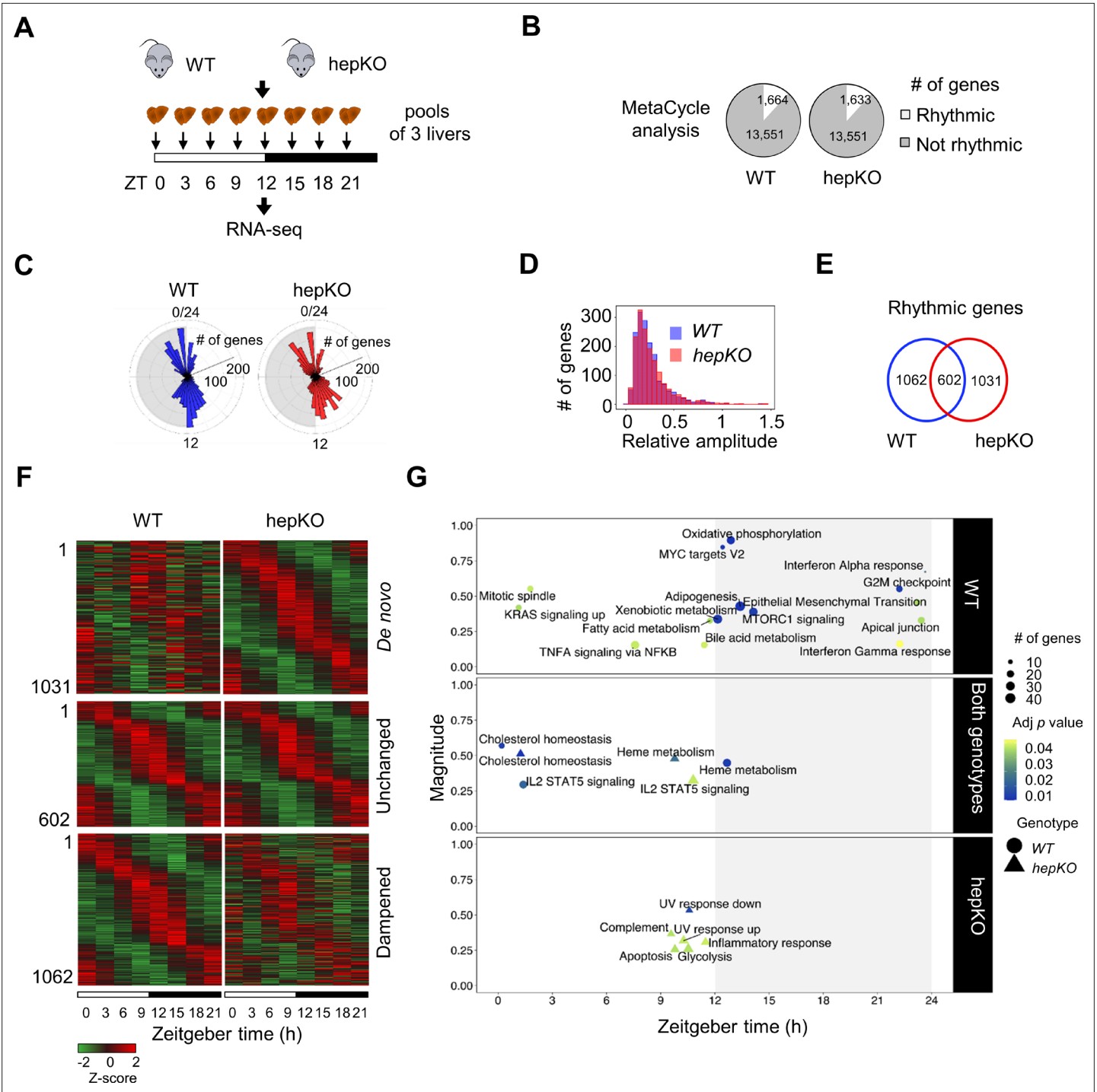

**Figure 2.** Deletion of hepatocyte KLF10 alters the circadian transcriptome in the liver. (**A**) Schematic illustrating the workflow used to assess the circadian transcriptomes of livers in WT and HepKO mice. (**B**) Number of rhythmic transcripts detected in the livers from WT and HepKO mice. (**C**) Phase distributions of rhythmic transcripts in the livers of WT and HepKO mice. (**D**) Relative amplitudes of rhythmic transcripts in the livers of WT and HepKO mice. (**E**) Number of unique and overlapping rhythmic transcripts in WT and HepKO mice. (**F**) Heatmaps of transcripts showing de novo, unchanged or dampened oscillatory behavior in the liver of HepKO mice compared to WT mice (pools of three livers per time point). (**G**) Magnitudes and phases of enriched biological processes identified by Phase Set Enrichment Analysis (PSEA) in WT mice only (top), in HepKO mice only (bottom), and in both genotypes (middle). Statistics: MetaCycle, significance threshold, p<0.05; PSEA; Kuiper test, significance threshold, $q < 0.01$.

The online version of this article includes the following source data and figure supplement(s) for figure 2:

**Source data 1.** MetaCycle analysis of the hepatic circadian transcriptome of WT and HepKO mice fed ad libitum and entrained in a 12:12 light/dark (LD)

*Figure 2 continued on next page*

*Figure 2 continued*

cycle.

**Source data 2.** Phase Set Enrichment Analysis (PSEA) of the hepatic transcriptome of WT and HepKO mice fed ad libitum and entrained in a 12:12 light/dark (LD) cycle.

**Figure supplement 1.** Deletion of hepatocyte KLF10 alters the circadian transcriptome in the liver.

**Figure supplement 1—source data 1.** Values and statistical test results for *Figure 2—figure supplement 1B-F*.

The alterations of biological pathways at the transcript level in hepKO mice prompted us to assess whether any changes associated with these pathways would be visible at the physiological level. Given the scope of our research question, we focused primarily on processes linked to glucose homeostasis. hepKO mice exhibited a modest increase of plasma glucose and a decrease in glycogen levels during their resting phase but not during the active phase (*Figure 2—figure supplement 1C and D*). The absence of hepatic KLF10 had no impact on the adaptation of HepKO mice to a 19 hr fast (glycemia: WT = 112 ± 7 mg/dl; hepKO = 113 ± 6 mg/dl). Additionally, we observed that hepKO hepatocytes displayed an increased glucose uptake and glucose production (*Figure 2—figure supplement 1E and F*). These observations suggest that the loss of KLF10 in hepatocytes leads to subtle physiological changes in carbohydrate processing at the cellular, tissue, and organism level. Together, these findings indicate that dysregulation of the hepatic circadian transcriptome in hepKO mice results in the rewiring of various pathways related to energy metabolism, which in turn may compromise the ability of these animals to adapt to dietary challenges.

## Hepatocyte KLF10 minimizes the adverse metabolic effects associated with a high sugar diet

Consumption of glucose and fructose in the form of sugar-sweetened beverages is linked to negative health outcomes (*Softic et al., 2020*). While previous studies have shown that glucose is a potent inducer of *Klf10* expression in primary hepatocytes (*Guillaumond et al., 2010*; *Iizuka et al., 2011*), the effect of fructose is unknown. We now show that primary hepatocytes treated with a mix of 5 mM (low) glucose and 5 mM fructose induces *Klf10* to the same extent as high (25 mM) glucose alone (*Figure 3*). Furthermore, treating cells with high glucose and 5 mM fructose has an additive effect (*Figure 3*).

Given these results in hepatocytes, combined with transcriptional changes occurring in unchallenged hepKO mice, we sought to better understand the role of KLF10 in mediating the effect of the two dietary sugars at the organism level. WT and hepKO mice were given ad libitum access to a chow diet with water (chow) or a chow diet with sugar-sweetened water (chow + SSW) for 8 weeks (*Figure 4A*). Similar to the response we observed in hepatocytes, the chow + SSW diet strongly induced hepatic *Klf10* mRNA expression when measured during the animals' feeding phase (ZT15) but not at the end of its fasting period—when *Klf10* is peaking (ZT9) (*Figure 4B*). Mice on the chow + SSW diet adapted their feeding behavior, with less calories coming from the chow pellets. As a result, their total calorie intake was unchanged relative to mice fed the chow diet (*Figure 4—figure supplement 1A*). After 8 weeks on the chow + SSW diet, mice from both genotypes displayed a significant increase in body mass and epididymal fat pad mass (*Figure 4C and D*). We also observed a trend for an increase in liver mass of mice fed a chow + SSW diet that reached significance in hepKO mice (*Figure 4E*). The chow + SSW diet increased liver triglycerides content, and histological signs of steatosis were evident in both genotypes; however, to a greater extent in hepKO mice (*Figure 4F*). Despite the greater increase in liver triglycerides in hepKO mice, no difference in circulating triglycerides was detected between genotypes (*Figure 4—figure supplement 1B*).

To further profile these mice, we assessed various parameters associated with sugar metabolism during the animals' early active (feeding) phase (ZT12-15)—a time of day when insulin sensitivity is at its maximum (*la Fleur et al., 2001*), and when pathways associated with nutrient intake and processing are activated (*Greenwell et al., 2019*; *Vollmers et al., 2009*). Glycemia was moderately increased in hepKO mice on chow diet as compared to WT mice fed the same diet (*Figure 4G*). Compared to WT mice, hepKO mice on the chow + SSW diet displayed hyperinsulinemia (*Figure 4H*). Furthermore, hepKO mice on the chow + SSW diet exhibited greater glucose intolerance compared to WT mice on the same diet (*Figure 4I*, *pAnova* [genotype:diet] = 0.00275). Next we profiled carbohydrate,

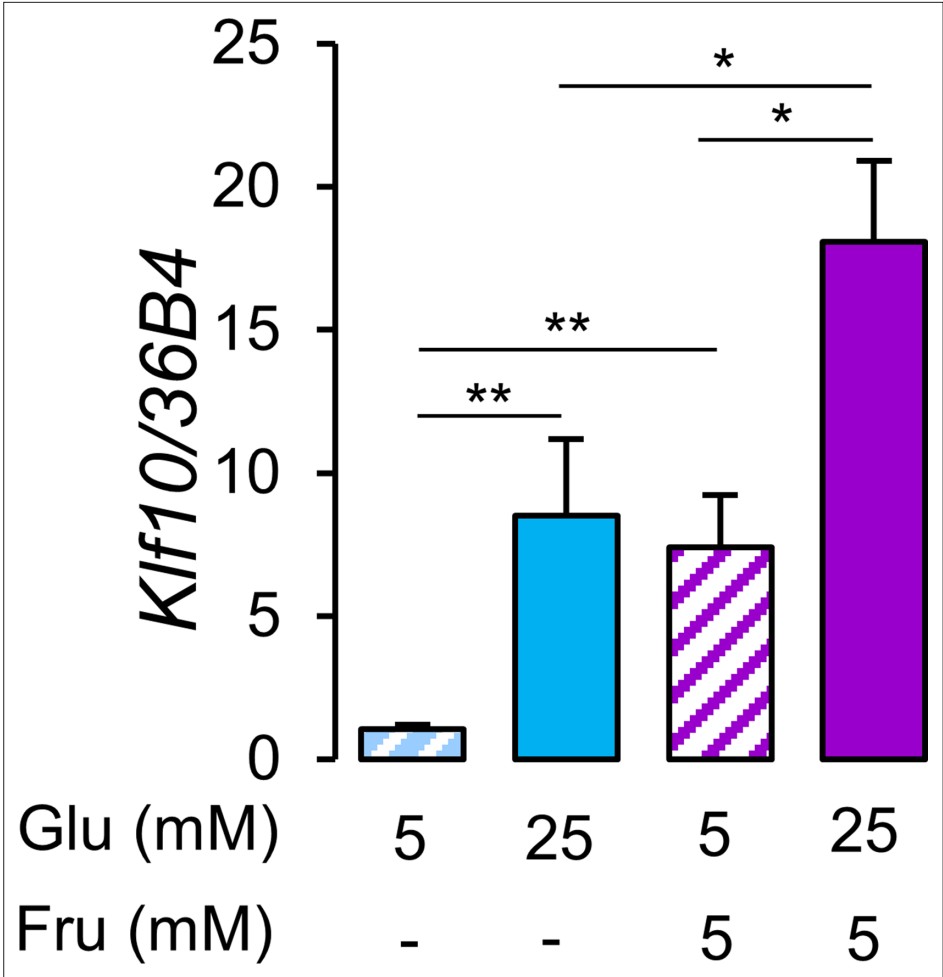

**Figure 3.** Glucose and fructose are potent inducers of *Klf10* expression in hepatocytes. Expression of the *Klf10* mRNA in primary WT mouse hepatocytes challenged with low (5 mM) or high (25 mM) glucose (Glu) in the absence or presence of 5 mM fructose (Fru) (mean ± SEM, n = 6) Statistics: nonparametric Kruskal–Wallis test. *p<0.05, **p<0.01.

The online version of this article includes the following figure supplement(s) for figure 3:

**Source data 1.** Values and statistical test results for *Figure 3*.

amino acid, and TCA-related metabolite levels in the four experimental groups of mice. We found that hexose-6-phosphate, myo-inositol, UDP-D-hexose, and citric acid were increased and that phospho-enol-pyruvate was decreased to a similar extent in both WT and hepKO mice on the chow + SSW diet relative to their respective counterpart fed a chow diet (*Figure 4J*). In contrast, glutamine was increased only in WT on the chow + SSW diet, while phenylalanine and fumarate were increased and decreased respectively only in hepKO mice on the chow + SSW diet (*Figure 4J*). These observations reveal subtle changes in amino acid catabolism in mice lacking hepatocyte KLF10.

We next assessed the effect of the chow + SSW diet at the transcriptional level in WT and hepKO mice by comparing the expression of various gene associated with metabolism. We detected an upregulation of *Slc2a4*—a gene encoding for a glucose transporter not normally expressed in the liver yet with well-described expression in adipose and skeletal muscle (*Klip et al., 2019*; *Figure 5A*). Furthermore, the absence of KLF10 in hepatocytes dramatically increased the expression of *Slc2a4* in hepKO mice on chow + SSW compared to WT mice on the same diet at both ZT9 and ZT15 (*Figure 5A*). The chow + SSW diet resulted in similar upregulation of glucose (*Slc2a2*) and fructose (*Slc2a5*) transporters in both WT and hepKO mice (*Figure 5—figure supplement 1A*). We found that the gene encoding for the rate-limiting glycolytic enzyme encoding *Pklr* was similarly upregulated by the chow + SSW challenge in both WT and HepKO mice at ZT9 while hepKO mice on the chow +

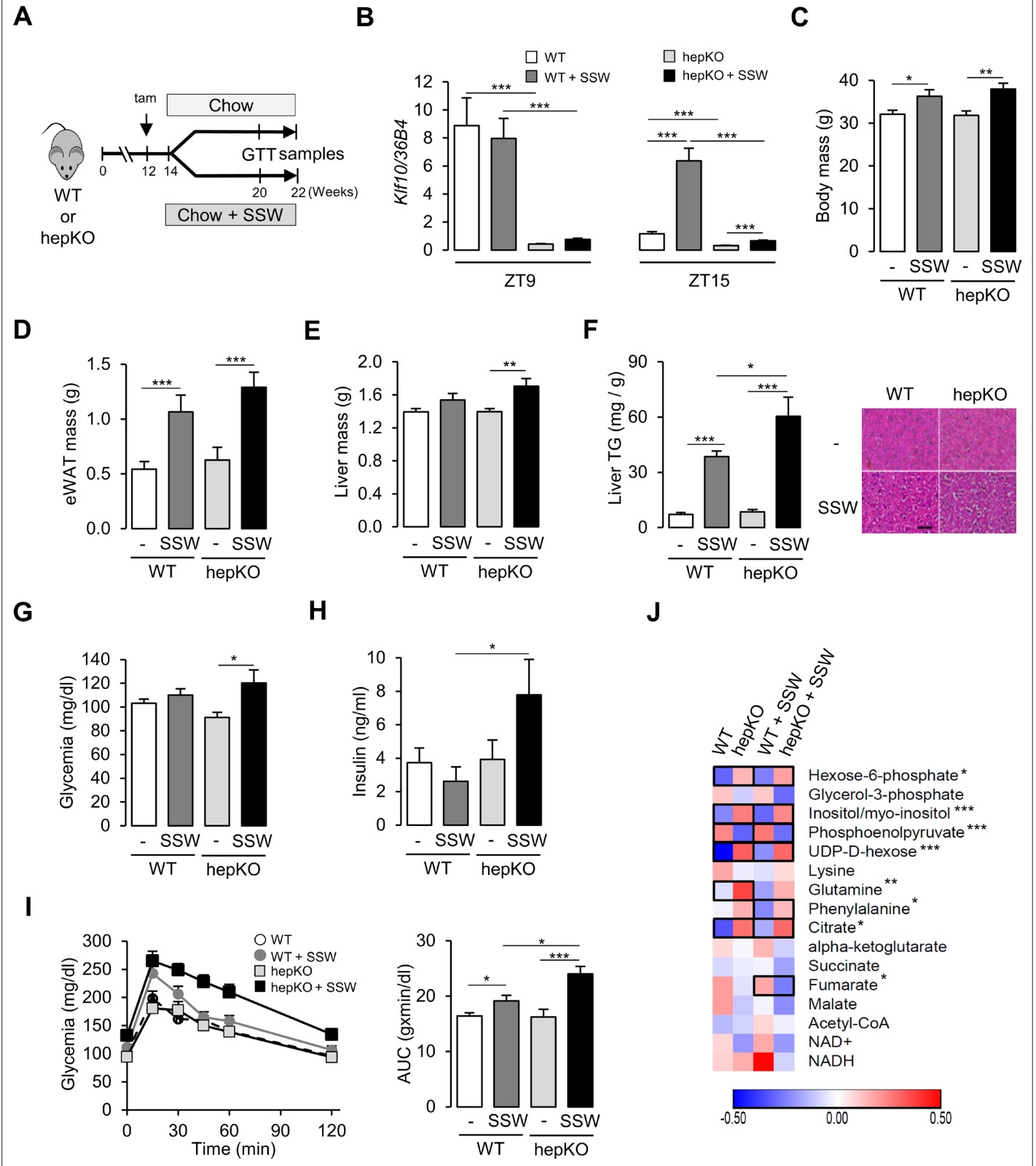

**Figure 4.** Loss of hepatocyte KLF10 exacerbates the adverse effects associated with increased sugar consumption. (**A**) Schematic illustrating the design of the sugar-sweetened water (SSW) challenge experiment. (**B**) Gene expression of *Klf10* at ZT9 and ZT15 in livers of WT and HepKO mice given a chow or chow + SSW diet (mean ± SEM, n = 4–8). (**C**) Body mass of mice given a chow or chow + SSW diet (mean ± SEM, n = 9–12). (**D**) Epididymal white adipose tissue mass of mice given a chow or chow + SSW diet (mean ± SEM, n = 9–12). (**E**) Liver mass of mice given a chow or chow + SSW diet (mean

*Figure 4 continued on next page*

*Figure 4 continued*

± SEM, n = 8–12). (**F**) Liver triglyceride content (mean ± SEM, n = 6–10) (left) and representative images of liver histology (right) in WT and HepKO mice given a chow or chow + SSW diet (scale bar = 50 µm). (**G**) Blood glucose in WT and HepKO mice given a chow or chow + SSW diet (n = 9–12). (**H**) Insulin levels in WT and HepKO mice given a chow or chow + SSW diet (mean ± SEM, n = 9–12). (**I**) Blood glucose levels assessed at regular intervals over a 2 hr period in mice undergoing a glucose tolerance test (GTT) performed at ZT12 (left) and area under the curve of blood glucose levels over the measurement period (right) (mean ± SEM, n = 9–12). (**J**) Heatmap showing the normalized concentration of metabolites in the liver of WT and HepKO mice given a chow or chow + SSW diet (mean, n = 9–12). Significant pairwise comparisons are boxed. (**C–H, J**), Measurements were performed during the animals' feeding period, at ZT15. Statistics: nonparametric Kruskal–Wallis test. *p<0.05, **p<0.01, ***p<0.005.

The online version of this article includes the following figure supplement(s) for figure 4:

**Source data 1.** Values and statistical test results for *Figure 3*.

**Figure supplement 1.** Loss of hepatocyte KLF10 exacerbates the adverse effects associated with increased sugar consumption.

SSW diet displayed higher expression of compared to WT mice on the same diet at ZT15 (*Figure 5A*). Expression of the gluconeogenic gene *Pck1*, a known direct KLF10 target gene (*Guillaumond et al., 2010*), was downregulated at ZT9 in WT mice on the chow + SSW diet as expected but was unchanged in hepKO mice on the same diet (*Figure 5B*). At ZT15, *Pck1* mRNA expression was only induced in hepKO mice on the chow + SSW diet (*Figure 5B*). No genotype effect was observed for glucokinase (*Gck*), ketohexokinase (*Khk*), and glucose-6-phosphatase (*G6pc*) (*Figure 5—figure supplement 1B*). Hepatocyte-specific deletion of KLF10 also altered the transcriptional response of genes involved in de novo lipogenesis as hepKO mice fed the chow + SSW diet displayed a significantly greater

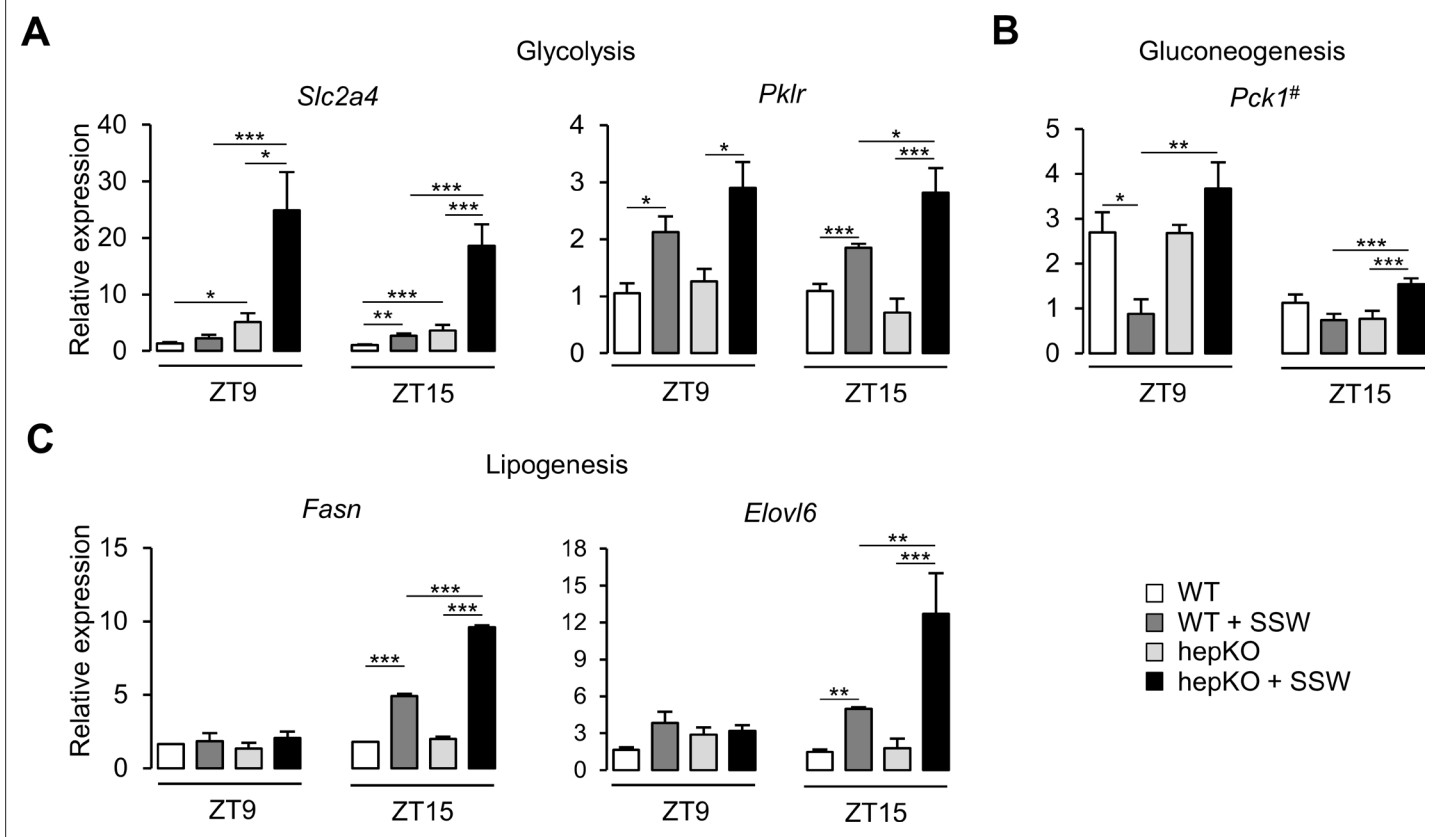

**Figure 5.** Altered metabolic gene expression in HepKO mice challenged with high sugar. (**A**) Expression profiles of *Slc2a4* and *Pklr*. (**B**) Expression profile of *Pck1*. (**C**) Expression profiles of the *Fasn* and *Elovl6*. (**A–C**) Expression was determined at ZT9 and ZT15 in the liver of WT and HepKO mice given a chow or chow + sugar-sweetened water (SSW) diet (mean ± SEM, n = 3–6). The # symbol denotes a direct KLF10 target. Statistics: nonparametric Kruskal–Wallis test. *p<0.05, **p<0.01, ***p<0.005.

The online version of this article includes the following figure supplement(s) for figure 5:

**Source data 1.** Values and statistical test results for *Figure 5*.

**Figure supplement 1.** Altered metabolic gene expression in HepKO mice challenged with high sugar.

induction of the genes encoding for fatty acid synthase (*Fasn*) and *fatty acid elongase* (*Elovl6*) at ZT15 as compared to their WT controls on the same diet (*Figure 5C*). Additionally, we observed a greater expression of ATP citrate lyase (*Acly*), acetyl-CoA carboxylase alpha (*Acaca*), malic enzyme (*Me1*), and thyroid hormone response SPOT14 (*Thrsp*) measured at ZT15 in hepKO mice on the chow + SSW diet as compared to WT mice on the same diet (*Figure 5—figure supplement 1C*).

Taken together, our physiological, metabolic, and molecular data suggests that the induction of KLF10 in response to a dietary sugar challenge serves as a protective mechanism that aids in safeguarding the organism from the detrimental effects associated with excess sugar intake.

## KLF10 is a 'transcriptional brake' that fine-tunes sugar signaling in hepatocytes

As the induction of hepatic KLF10 in chow + SSW -fed mice helps to minimize the adverse metabolic effects associated with excess dietary sugar intake, we hypothesized that this transcription factor may be involved in the metabolic adaptation of hepatocytes to acute changes in carbohydrate availability. To test this hypothesis in a cell-autonomous manner, we treated WT and hepKO primary hepatocytes with either 5 mM glucose (low glucose [LG]) or 25 mM glucose and 5 mM fructose (high glucose and fructose [HGF]) for 12 hr (*Figure 6A and B*) and performed RNA-seq to profile the transcriptomes under each condition.

The switch from the LG to the HGF condition led to a differential expression of 845 and 608 genes in WT and hepKO hepatocytes, respectively (*Figure 6C* and *Figure 6—source data 1*). Notably, the number of downregulated genes was decreased by ~40% in challenged hepKO hepatocytes compared to WT hepatocytes, suggesting that the ability of hepKO hepatocytes to repress gene expression in response to the HGF challenge is compromised (*Figure 6C*). Using the significantly altered transcripts detected under the LG and HGF conditions as an input, we performed pathway enrichment analysis and found several differences between WT and hepKO hepatocytes (*Figure 6C* and *Figure 6—source data 2*). In the LG condition, pathways associated with amino acid catabolism, fatty acid oxidation, and peroxisome metabolism were significantly enriched in WT but not hepKO hepatocytes (*Figure 6D*, left, and *Figure 6—source data 2*). Underlying these changes, WT hepatocytes displayed an upregulation of genes coding enzymes linked to branched chain amino acids (*Mccc2, Ivd, Acat1, Fah, Echs1, Adlh9a1*), alanine, lysine and tryptophan (*Echs1, Adlh9a1*), and tyrosine (*Fah*) degradation, as well as genes encoding for enzymes linked to mitochondrial and peroxisomal fatty acid oxidation (*Cpt1a, Echs1, Adlh9a1, Decr2*) (*Figure 6E and F*). In the HGF condition, hepKO hepatocytes displayed a greater number of enriched pathways relative to WT hepatocytes, with many of these de novo pathways associated with carbohydrate metabolism (*Figure 6D*, right). Similar to our results in mice challenged with the chow + SSW diet, the HGF condition resulted in an upregulation of the glucose transporter *Slc2a4* in the WT and hepKO hepatocytes, with the absence of KLF10 further augmenting this response (*Figure 6E*). In a similar manner, hepKO hepatocytes in the HGF condition displayed greater induction of genes associated with fructose (*Khk, Tfkc*) and glycogen (*Ppp1r3c*) metabolism relative to WT hepatocytes (*Figure 6D and E*). In addition to the enhanced upregulation of genes associated with carbohydrate metabolism, hepKO hepatocytes had a greater induction of genes associated with de novo lipogenesis (*Acly, Acaca, Fasn, Thrsp*) and long-chain fatty acid elongation (*Elovl6*) (*Figure 6E and F* and *Figure 6—figure supplement 1*).

Collectively, our in vitro challenge data supports the notion that KLF10 acts as a brake that aids in fine-tuning the hepatocyte's transcriptional response to glucose and fructose. Furthermore, the dysregulation of various metabolic pathways in hepatocytes lacking KLF10 corroborates the abnormal physiological response of hepKO mice challenged with the chow + SSW diet.

## KLF10 targets an extensive metabolic gene network in the liver

To further explore the role of KLF10 in the regulation of hepatic metabolism at the transcriptome level, we assessed the genome-wide occupancy of KLF10 in mouse liver using chromatin immunoprecipitation followed by high-throughput sequencing (ChIP-seq) (*Figure 7A*). We performed the analysis at ZT9, which is close to the protein acrophase (*Figure 7—figure supplement 1*). We identified a repertoire of 35,523 KLF10 binding sites matching the KLF10 matrix (*Schmitges et al., 2016*) and located in the –50/+ 1 kb region of associated genes (*Figure 7B*). The density of KLF10 binding sites was negatively correlated with their distance from the transcription initiation site with 37% being located in the

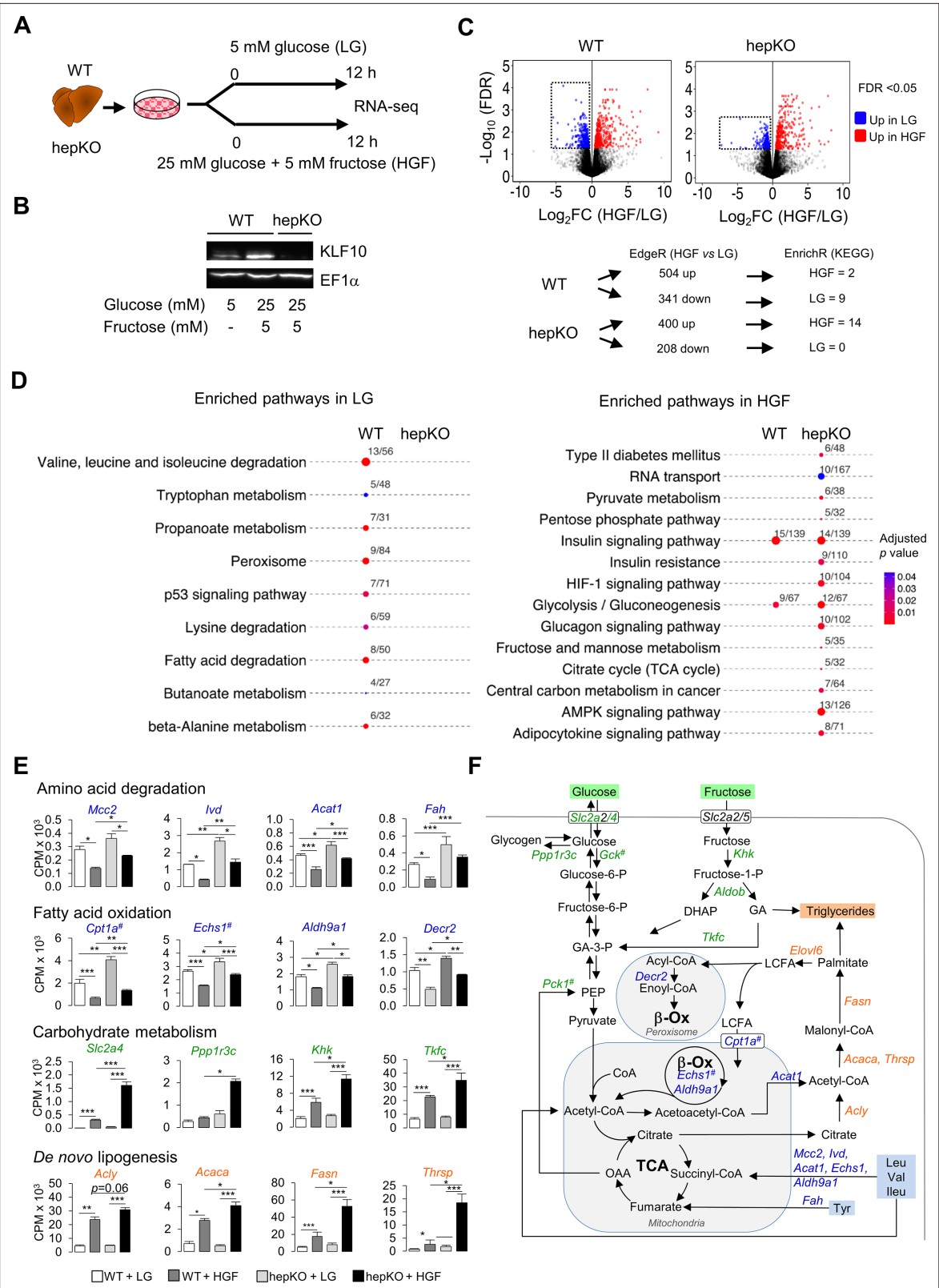

**Figure 6.** KLF10 governs the transcriptional response to hexose sugars in hepatocytes. (**A**) Schematic illustrating the design of the in vitro sugar challenge experiment in WT and HepKO hepatocytes treated with low glucose (LG) or high glucose and fructose (HGF). (**B**) Representative immunoblot showing KLF10 protein abundance in WT hepatocytes treated with LG or HGF and in HepKO primary hepatocytes treated with HGF. EF1α was used as loading control. (**C**) Top: volcano plots showing changes in gene expression in WT and HepKO hepatocytes treated with LG or HGF. Positive fold

*Figure 6 continued*

change (FC) values represent genes upregulated under HGF condition, and negative FC values represent genes upregulated under LG conditions. Genes significantly upregulated (FDR < 0.5) and displaying an FC > 2 in HGF (red dots) and in LG (blue dots) conditions. Dashed horizontal lines: FDR = 0.05; dashed vertical lines: FC = 2. Boxed areas highlight the larger number of downregulated genes in WT compared to HepKO hepatocytes. Bottom: summary of differential gene expression (DGE) data shown in the volcano plots and of the enrichment analysis. (D) Kyoto Encyclopedia of Genes and Genomes-enriched pathways in WT and HepKO primary hepatocytes treated with LG or HGF. (E) Expression profiles of genes associated with key metabolic pathways in WT and HepKO primary hepatocytes treated with LG or HGF (mean ± SEM, n = 3). (F) A schematic of key metabolic pathways in hepatocytes. Genes significantly upregulated in HepKO vs. WT hepatocytes when treated with HGF are highlighted on the schematic. DHAP: dihydoxyacetone phosphate; LCFA: long-chain fatty acids; GA: glyceraldehyde; OAA: oxaloacetic acid; PEP: phosphor-enol-pyruvate. In (E) and (F), the # symbol denotes a direct KLF10 target. Statistics: nonparametric Kruskal–Wallis test. *p<0.05; **p<0.01; ***p<0.005.

The online version of this article includes the following figure supplement(s) for figure 6:

**Source data 1.** Differentially expressed genes in WT and HepKO hepatocytes challenged with high sugar compared to unchallenged cells.

**Source data 2.** Kyoto Encyclopedia of Genes and Genomes-enriched pathway in WT and HepKO hepatocytes challenged with high sugar compared to unchallenged cells.

**Source data 3.** Values and statistical test results for *Figure 6E* and *Figure 6—figure supplement 1*.

**Figure supplement 1.** KLF10 governs the transcriptional response to hexose sugars in hepatocytes.

–10/+ 1 kb region, thus indicating that KLF10 mainly binds proximal promoter regions (*Figure 7B*). The 13,230 KLF10 binding sites present in the –10/+ 1 kb proximal region were associated to 7898 unique genes. To obtain a comprehensive view of the metabolic role of KLF10, we integrated in a single network the ChIP-seq data for these 7898 genes, the DGE data in hepKO vs. WT hepatocytes challenged with high sugar (HGF condition in *Figure 6*) and the HumanCyc metabolic pathways containing 6161 nodes including 2566 genes and 81,590 edges (*Romero et al., 2005*). The resulting KLF10 network visualized in Cytoscape containing 798 nodes and 10,980 edges revealed that genes regulated or/and bound by KLF10 were present in 23 annotated subnetworks (*Figure 7C*, *Figure 7—source data 1*, and *Figure 7—source data 2*). However, half of these KLF10 regulated/bound genes present in the network clustered in three major processes associated with lipid and amino acid metabolism as well as protein phosphorylation (*Figure 7C* and Cytoscape network supplement file). The intersect between differentially expressed genes in hepKO vs. WT HGF-treated hepatocytes and KLF10 bound genes revealed 93 direct targets (*Figure 7D*, top). This resulting list contained genes primarily associated with carboxylic acid metabolism, suggesting a role for KLF10 in central carbon metabolism (*Figure 7D*, bottom). In line with these observations, two of the top ranked direct target genes included *acyl-CoA synthetase short chain family member 2* (*Acss2*) and *acetyl-CoA carboxylase beta* (*Acacb*), which encode for two key enzymes of acetyl-CoA metabolism (*Figure 7—source data 1*). ChIP experiments targeting the KLF10 binding sites identified within the *Acss2* and *Acacb* genes using the ChIP-seq profiling experiment confirmed that both genes were bound by KLF10 at ZT15 in the liver of WT mice on chow + SSW diet (*Figure 7E*). Consistently, these two genes showed a greater upregulation at ZT15 in the liver of hepKO mice on chow + SSW diet compared to WT mice on the same diet, demonstrating that KLF10 is a transcriptional repressor of these two genes upon induction by high sugar during the feeding phase (*Figure 7F*). Interestingly, ACSS2 is one of the highly connected nodes in the KLF10 metabolic network, suggesting that KLF10-dependent repression of *Acss2* may indirectly impact multiple components of hepatic energy metabolism (*Figure 7—source data 1* and *Figure 7—source data 2*). We conclude from this data that KLF10 has an extensive repertoire of metabolic targets in the liver, many of which include key regulators of acetyl-CoA metabolism.

## Discussion

There is newfound appreciation of KLFs as transcriptional regulators of energy metabolism (*Hsieh et al., 2019*). This role is underscored by the identification of KLF variants involved in cardiometabolic disorders (*Oishi and Manabe, 2018*). In the liver, several of the *Klf* genes are direct targets of the CLOCK transcription factor (*Yoshitane et al., 2014*), suggesting that timed expression of these transcription factors may help facilitate the connection between the circadian clock and metabolism (*Jeyaraj et al., 2012*; *Panda, 2016*; *Reinke and Asher, 2019*; *Sinturel et al., 2020*).

In this study, we show that KLF10 is required for the circadian coordination of biological processes associated with energy metabolism in the liver while dispensable for an intact hepatic clock function.

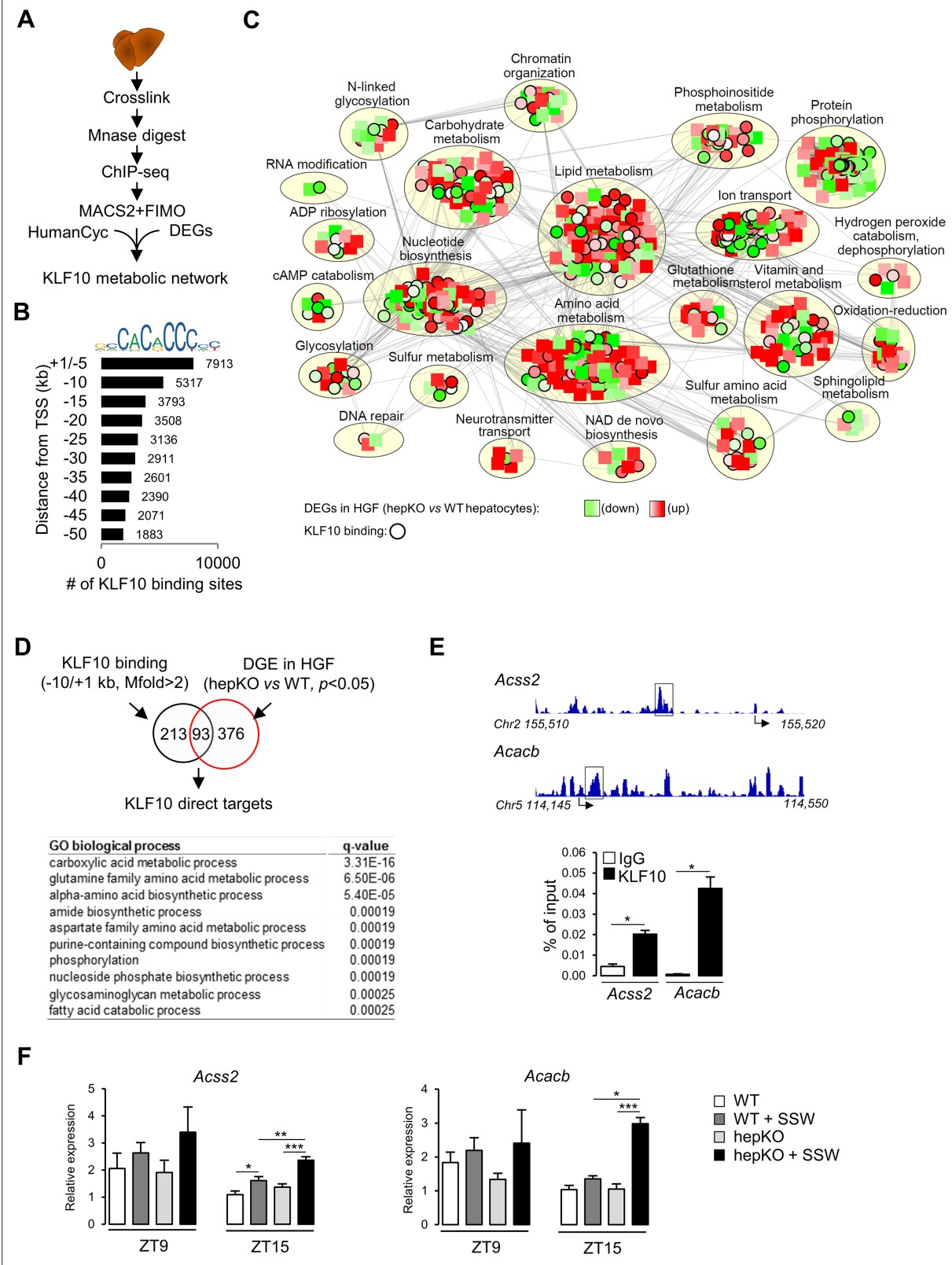

**Figure 7.** KLF10 regulates a large metabolic network in the liver. (**A**) Schematic illustrating the workflow used to identify KLF10 bound loci in mouse liver at ZT9. (**B**) Jaspar logo representing the KLF10 response element DNA sequence and distribution of KLF10 binding sites in mouse liver as a function of distance to the transcription start site. (**C**) Visualization of the KLF10 metabolic network constructed with Cytoscape using as inputs ChIP-seq data from mouse liver at ZT9, DGE data in high glucose and fructose (HGF)-treated hepatocytes and the HumanCyc metabolic network, (**D**) Venn diagram showing

*Figure 7 continued on next page*

*Figure 7 continued*

the number of direct KLF10 targets (top) and their associated GO biological process annotation (bottom). (**E**) ChIP-seq tracks near the *Acss2* and *Acacb* promoters (left) and ChIP of the boxed region at ZT15 in the liver of WT mice fed the chow + sugar-sweetened water (SSW) diet (right) (mean ± SEM, n = 4–6). (**F**) Expression of *Acss2* and *Acacb* at ZT9 and ZT15 in the liver of WT and HepKO mice given a chow or chow + SSW diet (mean ± SEM, n = 4–6). Statistics: nonparametric Kruskal–Wallis test. *p<0.05; **p<0.01; ***p<0.005.

The online version of this article includes the following figure supplement(s) for figure 7:

**Source data 1.** Genes bound by KLF10 in their –10/+1 kb region or differentially expressed in WT vs. HepKO hepatocytes challenged with high sugar.

**Source data 2.** Cytoscape file used to generate, visualize, and analyze the KLF10 regulated metabolic network .

**Source data 3.** Values and statistical test results for *Figure 7E and F*.

**Figure supplement 1.** KLF10 regulates a large metabolic network in liver.

This confirms our initial hypothesis that KLF10 acts primarily as a clock output (*Guillaumond et al., 2010*). Alteration of the rhythmic hepatic transcriptome in hepKO mice also led to de novo oscillation for a large number of genes, in a manner similar to the transcriptional changes occurring in KLF15-deficient cardiomyocytes (*Zhang et al., 2015*). In addition to the genetic disruption of clock-controlled KLFs, a variety of other systemic perturbations—such as high fat and ketogenic diets, arrhythmic feeding, caloric restriction, alcohol consumption, lung cancer, and rheumatoid arthritis—may result in de novo circadian oscillations in the liver (*Eckel-Mahan et al., 2013*; *Gaucher et al., 2019*; *Green-well et al., 2019*; *Makwana et al., 2019*; *Masri et al., 2016*; *Poolman et al., 2019*; *Tognini et al., 2017*). We therefore interpret the de novo oscillation seen in hepKO mice as the signature of their hepatic vulnerability as unchallenged hepKO mice appear healthy but are prone to develop hepatic steatosis associated with significant changes in metabolic gene expression when fed with high sugar. It is presently unknown whether the extensive reprogramming of the hepatic circadian transcriptome seen in many genetic or disease mouse models also occurs in humans. Should this occur, it may have important biomedical implications if diagnostic biomarkers or/and regulators of drug pharmacology that are expressed at low levels and arrhythmic in healthy subjects would become circadian upon disease initiation and progression, advocating for the integration of circadian timing in translational research (*Cederroth et al., 2019*).

The link between altered metabolic homeostasis and inflammation is well established (*Saltiel and Olefsky, 2017*). Recent studies have highlighted the role of KLF10 in regulating inflammation (*Huang et al., 2016*; *Wara et al., 2020*; *Yang et al., 2020*) and in attenuating liver injury in mice with NASH ad fibrosis (*Lee et al., 2020*). In this study, we found apoptosis and inflammation-related processes as gene sets that gain circadian coordination in unchallenged hepKO mice. Interestingly, inflammation markers also become rhythmic in a model of non-alcoholic steatohepatitis (NASH) (*Leclère et al., 2020*). The adverse metabolic phenotype displayed by hepKO mice on chow + SSW diet may therefore include an inflammatory component.

Hepatocyte KLF10 functions on several levels to protect the liver from the negative effects of excess dietary sugars. We found that it inhibits glucose uptake, and we obtained correlative evidence that this could be achieved by preventing the expression of the adipose and muscle-specific *Glut4* glucose transporter in hepatocytes. Previous studies have shown that *Klf10* is a glucose-responsive gene (*Guillaumond et al., 2010*; *Hirota et al., 2002*), a finding that together with our recent observation strongly suggests that KLF10 directs a negative feedback loop limiting excessive glucose uptake. Consistent with this role, glycolytic gene expression was upregulated in hepKO mice. In addition to glucose, we show that *Klf10* is induced by fructose, and the combination of these two hexose sugars amplifies this response. This upregulation is crucial in blunting the negative effects associated with sugar intake as hepKO mice challenged with a sugar-sweetened beverage containing glucose and fructose as exemplified by signs of insulin resistance and increased hepatic steatosis in these animals.

Our functional genomics data indicate that one role of KLF10 in liver is to repress the lipogenic gene network. During the metabolic adaptation of the liver to high sugar, the glucose-inducible factor carbohydrate-responsive element-binding protein (ChREBP) plays a critical role by inducing lipogenic genes (*Ma et al., 2006*; *Ortega-Prieto and Postic, 2019*). Interestingly, the induction of *Klf10* is also mediated by ChREBP (*Iizuka et al., 2011*). We therefore hypothesize a scenario in which sugar-induced steatosis is attenuated by KLF10, which acts both upstream of ChREBP—through repression of glucose uptake and downstream of ChREBP—via the repression of de novo lipogenesis.

Further, fructose is also converted by the gut microbiota to acetate which fuels ACLY-independent lipogenesis upon conversion to acetyl-coA by liver ACSS2 (*Zhao et al., 2020*; *Zhao et al., 2016*). Our findings indicate that KLF10 is a transcriptional repressor of *Acss2*, suggesting an additional mechanism used by KLF10 in aiding to suppress fructose-induced lipogenesis. Our findings are directly relevant to human health as the increased consumption of fructose in the form of both sweetened beverages and processed foods unarguably contributes to the current pandemic of obesity and NAFLD (*Hannou et al., 2018*; *Jensen et al., 2018*; *Mortera et al., 2019*). Links between the circadian clock and NAFLD have been suggested, but they are mostly based on preclinical models with disrupted core clock genes, a situation that is unlikely in humans (*Mazzoccoli et al., 2018*; *Mukherji et al., 2019*). The extensive rewiring of circadian gene expression and altered expression of clock-controlled regulators such as KLF10 associated with the onset or during disease states may disconnect circadian timing from metabolism and thereby contribute to chronic liver disease.

In summary, our work defines KLF10 as a dual transcriptional regulator with a circadian arm participating in the circadian coordination of liver metabolism and a homeostatic arm serving as a negative feedback control of sugar-induced lipogenesis. This duality is likely to be coordinated in healthy hepatocytes so that KLF10 peaks at the fasting/feeding transition when sugar metabolism shifts from gluconeogenesis to glycolysis. The desynchrony between circadian timing and the fasting/feeding cycle presumably impairs KLF10 functionality and thus may further exacerbate metabolic disease phenotype.

# Materials and methods

## Key resources table

| Reagent type (species) or resource | Designation | Source or reference | Identifiers | Additional information |
|---|---|---|---|---|
| Genetic reagent (*Mus musculus*, C57BL/6) | WT | *Weng et al., 2017* | | Klf10flox/flox controls; used males for experiments |
| Genetic reagent (*Mus musculus*, C57BL/6) | *Alb-CreER^T2* | *Schuler et al., 2004* | | |
| Genetic reagent (*Mus musculus*, C57BL/6) | hepKO | This study | | Used males for experiments |
| Cell line (*Mus musculus*) | WT | This study | | Primary heaptocytes from males; used 150,000–250,000 cells per plate as indicated |
| Cell line (*Mus musculus*) | hepKO | This study | | Primary hepatocytes from males; used 150,000–250,000 cells per plate as indicated |
| Biological sample (*Mus musculus*) | Liver, heart, kidney, spleen, lung, white adipose tissue | This study | | |
| Biological sample (*Mus musculus*) | Blood | This study | | |
| Antibody | Anti-KLF10 (mouse monoclonal) | CDI Laboratories | Cat. #m14-355 | WB (1:800) |
| Antibody | Anti-EGF1a (mouse monoclonal) | Upstate signaling solutions | Cat. #05-235 | WB (1:1000) |
| Antibody | Anti-IgG (goat polyclonal) | Sigma | Cat. #A4416 | WB (1:40,000) |
| Commercial assay or kit | Glucose Hexokinase Assay Kit | Sigma | Cat. #GAHK20-1KT | |
| Commercial assay or kit | PowerUp SYBR green Master Mix | Applied Biosystems | Cat. #A25779 | |
| Commercial assay or kit | QIAquick PCR Purification Kit | Qiagen | Cat. #2810 | |
| Commercial assay or kit | Triglycerides colorimetric assay kit | Cayman | Cat. #10010303 | |
| Commercial assay or kit | NucleoSpin RNA | Macherey-Nagel | Cat. #740955250 | |
| Commercial assay or kit | RNeasy mini kit | QIAGEN | Cat. #74104 | |

| Reagent type (species) or resource | Designation | Source or reference | Identifiers | Additional information |
|---|---|---|---|---|
| Commercial assay or kit | NEBNext Ultra RNA Library Prep Kit for Illumina | New England Biolabs | Cat. #E7530S | |
| Commercial assay or kit | QubitTM dsDNA HS Assay Kit | Invitrogen | Cat. #Q32854 | |
| Commercial assay or kit | Glucose Uptake-Glo Assay | Promega | Cat. #J1341 | |
| Commercial assay or kit | Mouse Insulin Elisa | Mercodia | Cat. #10-1247-01 | |
| Commercial assay or kit | Glucose (GOD-PAP) | Randox | Cat. #8318 | |
| Chemical compound, drug | DMEM, low glucose, GlutaMAX Supplement, pyruvate | Gibco | Cat. #21885025 | |
| Chemical compound, drug | DMEM, high glucose, GlutaMAX Supplement, pyruvate | Gibco | Cat. #10569010 | |
| Chemical compound, drug | DMEM, glucose-free | Gibco | Cat. #11966025 | |
| Chemical compound, drug | William's E medium | Gibco | Cat. #22551022 | |
| Chemical compound, drug | Dexamethasone | Sigma | Cat. #D0700000 | Used 10 nM final concentration |
| Chemical compound, drug | Insulin-selenium-transferrin mix | Gibco | Cat. #41400045 | Used 1× final concentration |
| Chemical compound, drug | Insulin (human, recombinant zinc) | Thermo Fisher Scientific | Cat. #12585014 | Used 860 nM final concentration |
| Chemical compound, drug | Percoll | GE Healthcare | Cat. #17089101 | |
| Chemical compound, drug | Protein G Dynabeads | Thermo Fisher Scientific | Cat. #10003D | |
| Chemical compound, drug | Complete protease inhibitor cocktail | Roche | Cat. #11836153001 | |
| Chemical compound, drug | Collagenase CLS3 | Worthington | Cat. #WOLS04180 | Used 500 U/mL final concentration |
| Chemical compound, drug | Collagenase CLS4 | Worthington | Cat. #WOLS04186 | Used 500 U/mL final concentration |
| Chemical compound, drug | Amyloglucosidase | Sigma | Cat. #11202332001 | Used 2 mg/mL final concentration |
| Chemical compound, drug | SuperScript II Reverse Transcriptase | Invitrogen | Cat. #18064014 | Used 200 U per reaction |
| Chemical compound, drug | Micrococcal nuclease | Thermo Fisher Scientific | Cat. #88216 | Used 80 U per reaction |
| Chemical compound, drug | Protein G Dynabeads | Thermo Fisher Scientific | Cat. #10003D | |
| Software, algorithm | R | Bioconductor | https://www.bioconductor.org/ | |
| Software, algorithm | Trimmomatic | *Bolger et al., 2014* | http://www.usadellab.org/cms/ | |
| Software, algorithm | FastQC | Babraham Institute | https://www.bioinformatics.babraham.ac.uk/projects/fastqc/ | |
| Software, algorithm | FASTQ Groomer | Galaxy project | https://sourceforge.net/projects/fastqgroomer/ | |
| Software, algorithm | Bowtie2 | *Langmead and Salzberg, 2012* | https://github.com/BenLangmead/bowtie2 | RRID: SCR_016368 |
| Software, algorithm | BEDtools | *Quinlan and Hall, 2010* | https://github.com/arq5x/bedtools2 | RRID: SCR_006646 |
| Software, algorithm | Rsubread (featureCounts) | *Liao et al., 2014* | https://git.bioconductor.org/packages/Rsubread | RRID: SCR_009803 |
| Software, algorithm | MetaCycle | *Wu et al., 2016* | https://github.com/gangwug/MetaCycleApp | |
| Software, algorithm | Phase Set Enrichment Analysis | *Zhang et al., 2016* | https://github.com/ranafi/PSEA | |
| Software, algorithm | MACS2 | *Feng et al., 2012* | https://pypi.org/project/MACS2/ | |

*Continued on next page*

*Continued*

| Reagent type (species) or resource | Designation | Source or reference | Identifiers | Additional information |
| --- | --- | --- | --- | --- |
| Software, algorithm | GREAT | *McLean et al., 2010* | http://great.stanford.edu/public/html/splash.php | |
| Software, algorithm | FIMO | *Bailey et al., 2015* | http://meme-suite.org/tools/fimo | |
| Software, algorithm | FastHeinz | *Beisser et al., 2010* | https://www.bioconductor.org/packages/release/bioc/html/BioNet.html | |
| Software, algorithm | clusterMaker2 | *Morris et al., 2011* | http://www.rbvi.ucsf.edu/cytoscape/clusterMaker2/ | |
| Software, algorithm | Bingo | *Maere et al., 2005* | https://github.com/cytoscape/BiNGO | |
| Software, algorithm | Cytoscape | *Shannon et al., 2003* | https://cytoscape.org/ | |
| Other | Qubit fluorometer | Thermo Fisher Scientific | Cat. #Q33238 | |
| Other | FreeStyle InsuLinx glucometer | Abbott | N/A | |
| Other | Laboratory rodent chow diet | Safe diets | Cat. #R03-25 | |
| Other | Apistar syrup (organic) | ICKO | Cat. #HC406 | |

## Mouse models, housing, and diets

To generate a hepatocyte-specific conditional *Klf10* null allele, mice in the C57BL/6 background and bearing a *Klf10* exon1 flanked by LoxP sites (*Klf10^{flox/flox}*) described previously (*Weng et al., 2017*) were crossed with a serum albumin-driven *Cre-ER^{T2}* mouse line (*Schuler et al., 2004*). To obtain mice with a loss of KLF10 in hepatocytes (HepKO), 5- week-old male *Klf10^{flox/flox}*; *Alb^{+/CreERT2}* mice were gavaged with tamoxifen (10 mg/kg body weight in 200 µl sunflower oil). For all experiments, aged-matched, tamoxifen-treated, *Klf10^{flox/flox}* male mice were used as wild-type controls (WT). Genotyping was performed by PCR using the primers listed in *Supplementary file 1*.

Mice were housed in a temperature- and humidity-controlled room (21°C ± 2 °C/70%), maintained on a 12 hr:12 hr LD cycle and fed a standard laboratory chow diet (SafeDiets A03) ad libitum. For the in vivo sugar challenge, mice from each genotype were randomized between the experimental groups and had ad libitum access to standard laboratory chow diet and either water or SSW beverage (water +30% Apistar solution [35% fructose, 33% glucose, 32% sucrose]) for 8 weeks. One animal that did not gain weight after 8 weeks was excluded.

All animal studies were approved by the local committee for animal ethics Comité Institutionnel d'Éthique Pour l'Animal de Laboratoire (CIEPAL-Azur; authorized protocols: PEA 244 and 557) and conducted in accordance with the CNRS and INSERM institutional guidelines.

## Primary hepatocyte isolation and culture conditions

Primary hepatocytes were isolated using a retrograde in situ perfusion procedure as described previously (*Mederacke et al., 2015*) with a few modifications. Briefly, mice were anesthetized between ZT3-ZT6, and upon cannulation via the inferior vena cava, the liver was perfused at a flow rate of 6 mL/min with a warmed (37 °C), EGTA solution. After 3 min, the buffer was exchanged for a glucose- and EGTA-free buffer containing 45 mM $CaCl_2$, collagenase types CLS3 (500 U/mL) and CLS4 (500 U/mL) (Worthington Biochemical Corporation), and the liver was perfused in this buffer for an additional 5 min at a flow rate of 6 mL/min. After perfusion, the liver was excised and placed in Leibovitz L-15 medium supplemented with 10% fetal calf serum (FCS), 1% penicillin and streptomycin (P/S). Hepatocytes were released by mechanical disruption and filtered through a 70 µm cell strainer and resuspended in 50 mL of Leibovitz L-15, 10% FCS, 1% P/S medium. To seed hepatocytes, the solution was centrifuged at 400 rpm for 2 min and hepatocytes were resuspended in 8 mL William's E medium supplemented with 860 nM insulin, 1% P/S, 10% FCS, and 1% glutamate. Dead cells were removed by Percoll density-gradient centrifugation. Unless otherwise indicated, hepatocytes were seeded in collagen-coated 6-well plates at a density of 250,000 cells/plate. After hepatocyte attachment, medium was substituted for low-glucose DMEM, supplemented with GlutaMAX, pyruvate, 10 nM

dexamethasone (Dex), and 1× insulin-selenium-transferrin (ITS). All hepatocyte manipulations took place the following day, ~12–15 hr after cell attachment.

## Food-seeking behavioral analysis

Mice were individually housed with ad libitum access to food and water. An infrared detection system (Actimetrics) was used to assess food seeking behavior. Mice were weighed weekly and their daily food intake was estimated by measuring the difference between the quantity of food provided and food remaining after 1 week. Actogram plots were generated using the ImageJ plugin 'ActogramJ' (*Schmid et al., 2011*).

## Blood analysis

Unless stated otherwise, all measurements were performed at ZT15. Glycemia levels were measured using a FreeStyle Papillon glucometer (Abbott). Plasma triglycerides were measured using a colorimetric assay kit (Cayman) according to the manufacturer's protocol. Plasma insulin was measured using a Mouse Insulin Elisa kit (Mercodia) in accordance with the manufacturer's protocol.

## Glycogen assay

Hepatic glycogen was measured as described previously (*Feillet et al., 2016*). Hepatic glycogen was extracted from samples collected at ZT3 and ZT15. Upon precipitation, the glycogen pellet was either resuspended in 2 mg/mL amyloglucosidase (Sigma-Aldrich) or in 0.2 M sodium acetate (pH 4.9) to measure total glucose and free glucose, respectively. Total and free glucose was assayed using a glucose-hexokinase assay kit (Sigma-Aldrich) according to the manufacturer's protocol.

## Glucose tolerance test

Mice were fasted at ZT6 for 6 hr and then injected intraperitoneally with glucose (0.5 g/kg body weight) in 1× PBS. Blood glucose was measured from tail-blood microsamples (<0.5 µL) just before injection and at regular intervals over 2 hr using a FreeStyle Papillon glucometer (Abbott).

## Liver triglycerides

Total lipids were extracted from liver samples collected at ZT15 using a modified Folch method (*Folch et al., 1957*). Briefly, 50–100 mg of liver were homogenized in a methanol/chloroform (2:1, v/v) solution. After addition of 1 volume of chloroform and 0.9 volume of water followed by mixing, the organic phase was separated, evaporated, and resuspended in NP40. Triglycerides content was measured using a colorimetric assay kit (Cayman) according to the manufacturer's protocol.

## Histology

Liver tissue was excised, fixed with 4% PFA solution overnight, processed for paraffin embedding, sectioned at 5 µm thickness, and then stained for hematoxylin and eosin (H&E). Liver sections were then imaged using the Vectra Polaris digital slide scanner (CLS143455, Akoya Biosciences Inc) and subjected to blind scoring.

## Glucose uptake

Primary hepatocytes from WT and HepKO were seeded in collagen-coated 24-well plates at a density of 150,000 cells per well and incubated overnight in DMEM containing 25 mM glucose, 17 nM ITS, and 10 nM Dex. The following day, cells were washed twice with 1× PBS before stimulation with a glucose-free DMEM containing 17 nM ITS and 1 µM 2-deoxyglucose (2-DG). Hepatocytes were left in this medium for 10 min before the reaction was stopped. Lysates were then assayed for glucose uptake using the Glucose Uptake-Glo Assay kit (Promega) as per the manufacturer's protocol. Relative light unit (RLU) values were normalized to the total protein concentration of each corresponding sample.

## Glucose production

Primary hepatocytes from WT and HepKO were isolated, seeded in collagen-coated 6-well plates at a density of 250,000 cells per well, and incubated overnight in DMEM containing 5 mM glucose, 17 nM ITS, and 10 nM Dex. The following day, the media was removed and hepatocytes were given

glucose-free DMEM supplemented with 17 nM ITS and 10 nM Dex, 16 mM lactate, and 4 mM pyruvate. Media (250 µL) was collected 24 hr after replacement. The amount of glucose in each sample was determined using a GOD-PAP colorimetric assay (Randox) as per the manufacturer's recommendations.

## Liver extract metabolite identification

Liver metabolites were extracted following a protocol adapted from *Hui et al., 2017*. Briefly, 10 µL/mg liver of –20 °C methanol:acetonitrile:water (40:40:20, v/v/v) solution was added to approximately 50 mg of liver. The mixture was crushed using a pellet pestle for 30 s, followed by sonication for 5 min and vortexing for 15 s. Samples were incubated at 4 °C for 10 min to precipitate proteins before centrifugation at 4 °C and 15,000 *g* for 10 min. The supernatant was transferred to LC-MS vials for analyses.

Liver crude organic extracts were analyzed using a Vanquish UHPLC coupled with a Thermo Q-Exactive (Thermo Fisher Scientific GmbH, Bremen, Germany) mass spectrometer (MS) and an ESI source operated with Xcalibur (version 2.2, Thermo Fisher Scientific) software package. Metabolites separation was achieved on a Waters Acquity BEH C8 (1 × 150 mm, 1.7 µm) column with an injection volume of 5 µL and a flow rate of 0.1 mL/min. The mobile phase was composed of octylammonium acetate 4 mM in $H_2O$ adjusted to pH = 4.6 with acetic acid (phase A) and methanol (phase B) and the following gradient was used: 1% B for 5 min, 1–95% B in 10 min, 95–99% B in 5 min, 99% B held for 3 min, then 1% B for 7 min for a total run time of 30 min. Data acquisition was realized under full scan switch (positive and negative) mode ionization from *m/z* 80–900 at 70,000 resolution followed by inclusion list-dependent MS/MS. Samples were randomized and analyzed in triplicates. A quality control (QC) sample was added by pooling equal volumes of each sample to monitor suitability, repeatability, and stability of the system and injected every 10 samples. An extraction blank sample was also injected every 10 runs in order to monitor sample cross-contamination. Pure commercial standards (fumaric acid, succinic acid, oxaloacetic acid, alpha-ketoglutaric acid, glutamic acid, phosphoenolpyruvic acid, dihydroxy acetone phosphate, citric acid, NADHP, and pyruvic acid, Merck) were analyzed to monitor their retention time, *m/z* of the parent ion and fragment ions. Feature extraction, compound identification, and retention time correction were performed using Compound Discoverer 2.1 software (Thermo Fisher Scientific). Processing of the filtered feature table was realized using Metaboanalyst (http://www.metaboanalyst.ca; *Chong et al., 2019*). Samples were normalized using QCs, data were log transformed then Pareto-scaled (mean-centered and divided by the square root of the standard deviation of each variable). The QCs were then removed from the normalized data table to perform statistical analyses.

## In vitro stimulation of hepatocytes with glucose and fructose

After seeding of WT and HepKO hepatocytes, medium was replaced with DMEM supplemented with GlutaMAX, pyruvate, 10 nM Dex, 1× ITS and containing either (1) 5 mM glucose, (2) 25 mM glucose, (3) 5 mM glucose and 5 mM fructose, or (4) 25 mM glucose and 5 mM fructose. After a 12 hr incubation period, media was aspirated and hepatocytes were lysed with Qiagen RLT buffer. Lysed material containing RNA was collected, flash frozen, and stored at –80 °C until RNA extraction.

## RNA isolation and quantitative real-time PCR

Total RNA was purified using RNeasy extraction mini kit (Qiagen) as per the manufacturer's instructions. To generate cDNAs, 1 µg of total RNA was reverse transcribed using random hexamer primers and Superscript II reverse transcriptase (Invitrogen). Diluted cDNAs (1/10) were added to a PowerUp SYBR green master mix (Applied Biosystems) and amplified using a StepOnePlus Real-Time PCR system (Applied Biosystems) as per the manufacturer's instructions. The relative mRNA abundance was calculated using the ΔCt method and normalized to the value of the expression of the housekeeping gene *36B4*. The sequences of primers used are listed in *Supplementary file 1*.

## Western blotting

Whole-cell extracts from liver tissue or isolated hepatocytes were processed using the Active Motif Nuclear Extraction kit as per the manufacturer's instructions. Lysates were separated by SDS–PAGE and transferred to PVDF membranes (Bio-Rad). Immunoreactive protein was detected using ECF reagent (Millipore) and Fusion FX7 (Vilber). The primary antibodies used were human anti-KLF10 mAb

(1:800) and mouse anti-EGF1a mAb (1:1000). The secondary antibody used was goat anti-mouse IgG (1:40,000).

## RNA-sequencing

For the circadian time-course experiment, equal amounts of total RNA extracted from three mouse livers were pooled at each of the eight time points assessed over a 24 hr period. Preparation of libraries and sequencing was performed at UCAGenomiX (Sophia-Antipolis, France). Here, libraries were prepared using the Truseq Stranded Total RNA Preparation Kit with Ribo-Zero Gold (Illumina) following the manufacturer's recommendations. Normalized libraries were multiplexed, loaded on a flow cell (Illumina), and sequenced on a NextSeq 500 platform (Illumina) using a 2 × 75 bp paired-end (PE) configuration. Image analysis and base calling were conducted by the HiSeq Control Software (HCS). Raw sequencing data (.bcl files) generated from Illumina NextSeq 500 platform was converted into fastq files and de-multiplexed using Illumina's bcl2fastq 2.17 software.

For the in vitro sugar challenge experiment, RNA was extracted from WT and HepKO hepatocytes treated with either 5 mM glucose or 25 mM glucose and 5 mM fructose for 12 hr (three biological replicates per experimental condition). Preparation of libraries and sequencing was performed at GENEWIZ (Leipzig, Germany). Here, libraries were prepared using NEBNext Ultra RNA Library Preparation Kit for Illumina (New England Biosystems) following the manufacturer's recommendations. Normalized libraries were multiplexed, loaded on a flow cell (Illumina), and sequenced on a HiSeq 4000 platform (Illumina) using a 2 × 150 PE configuration. Image analysis and base calling were conducted by the HiSeq Control Software. Raw sequencing data (.bcl files) generated from Illumina HiSeq 4000 platform was converted into fastq files and de-multiplexed using Illumina's bcl2fastq 2.17 software.

## Chromatin immunoprecipitation

Snap frozen liver (~300 mg) was cross-linked with 1.1% formaldehyde in PBS for 20 min at room temperature and cross-linking was quenched with 125 mM glycine for 5 min. After Dounce homogenization, the crushed tissue was washed twice in cold PBS, resuspended in cold cell lysis buffer (50 mM HEPES [pH 8], 200 mM NaCl, 20 mM EDTA, 12% glycerol, 1% NP40, 0.8% Triton-X100, 1 mM PMSF, protease inhibitor cocktail), and sonicated 3× (15 s on/30 s off cycle, low-power mode) in a Bioruptor (Diagenode). Nuclei were purified by sedimentation at 1000 rpm for 5 min, the supernatant was removed, and the pellet was resuspended in nuclei preparation buffer (0.34 M sucrose, 15 mM Tris [pH 8], 15 mM KCl, 0.2 mM EDTA, 0.2 mM EGTA, protease inhibitor cocktail). To digest chromatin, the suspension was incubated with 80 Units micrococcal nuclease (MNase) for 10 min at 37 °C. Next, the suspension was spun at 6600 rpm for 5 min, the supernatant was discarded, and nuclei were resuspended in nuclei lysis buffer (50 mM Tris [pH 8], 150 mM NaCl, 10 mM EDTA, 0.5% NP40, 1% Triton-X100, 0.4% SDS, and protease inhibitor cocktail). The sample was sonicated for 12× (30 s on/30 s off, high-power mode) in a bioruptor to release digested chromatin. The suspension containing the fragmented chromatin was centrifuged at 10,000 rpm for 8 min at 4 °C to remove debris and the supernatant was diluted in ChIP dilution buffer (50 mM Tris pH 8, 150 mM NaCl, 1% Triton-X100, protease inhibitor cocktail). Protein-DNA complexes were immunoprecipitated using 3–4 μg of human anti-KLF10 mAb (CDI Laboratories) incubated at 4 °C overnight and subsequently isolated with protein G Dynabeads for 2 hr at room temperature. Beads were washed four times in low-salt buffer (50 mM Tris [pH 7.4], 5 mM EDTA, 150 mM NaCl, 1% Triton X-100, 0.5% NP40), four times in high-salt wash buffer (100 mM Tris [pH 7.4], 250 mM NaCl, 1% NP40, 1% sodium deoxycholate), once in TE buffer (50 mM Tris [pH 7.4], 10 mM EDTA), and then eluted with 1% SDS-0.1 M NaHCO$_3$. Reverse crosslinking of protein-DNA complexes was performed overnight by incubation at 65 °C followed by proteinase K digestion for 4 hr at 42 °C. DNA fragments were purified using QIAquick PCR purification columns (Qiagen) quantified using a Qubit fluorometer (Invitrogen). Fragmented DNA was flash frozen and stored at –80 °C until further processing for high-throughput sequencing or qRT-PCR.

## ChIP-sequencing

Preparation of libraries and sequencing was performed at GENEWIZ (Leipzig, Germany). Here, libraries were prepared using NEBNext Ultra DNA Library Preparation Kit for Illumina (New England Biosystems) following the manufacturer's recommendations. Briefly, the ChIP DNA was end repaired

and adapters were ligated after adenylation of the 3′ ends. Adapter-ligated DNA was size selected, followed by clean up, PCR enrichment. ChIP libraries were validated using an Agilent TapeStation and quantified using Qubit 2.0 Fluorometer as well as qRT-PCR (Applied Biosystems, Carlsbad, CA, USA). Normalized libraries were multiplexed, clustered, and loaded on one lane of a flow cell (Illumina). Libraries were sequenced on a HiSeq 4000 platform (Illumina) using a 2 × 150 PE configuration. Image analysis and base calling were conducted by the HiSeq Control Software (HCS). Raw sequence data (.bcl files) generated from Illumina HiSeq was converted into fastq files and de-multiplexed using Illumina's bcl2fastq 2.17 software. One mismatch was allowed for index sequence identification.

### Network analysis

A Network integrating RNA-seq data from the HGF condition of the in vitro sugar challenge experiment and ChIP-seq data was generated using FastHeinz from the R BioNet package (*Beisser et al., 2010*) with the HumanCyc metabolic pathways (http://humancyc.org/) downloaded from http://www.ndexbio.org (*Pillich et al., 2017*). Next, a weight matrix was constructed with all expressed genes in the WT and HepKO primary hepatocytes taking the lowest p-value from the ChIP-seq RNA-seq datasets. Network visualization was done using Cytoscape and AutoAnnotate app (*Kucera et al., 2016*; *Shannon et al., 2003*). Clustering was performed using the MCL partitioning algorithm from clusterMaker2 (*Morris et al., 2011*), and gene ontology analysis was performed using BiNGO (*Maere et al., 2005*) at http://impala.molgen.mpg.de/ (*Kamburov et al., 2011*) and at https://www.uniprot.org/.

### Alignment, quantification, and statistical analyses

For the circadian time-course experiment, sequences were preprocessed using FASTQ Groomer and checked using FastQC (https://www.bioinformatics.babraham.ac.uk/projects/fastqc/). Reads were mapped against the *Mus musculus* genome (mm10) downloaded from Ensembl.org using STAR_2.4.0a (*Dobin et al., 2013*) with the Encode RNA-seq best practices options indicated ("–outFilterIntron-MotifsRemoveNoncanonicalUnannotated –alignMatesGapMax 1000000 –outReadsUnmapped Fastx –alignIntronMin 20 –alignIntronMax 1000000 –alignSJoverhangMin 8 –alignSJDBoverhangMin 1 – outFilterMultimapNmax 20"). After alignment to the genome reads were counted with featureCounts (*Liao et al., 2014*) with "–primary –g gene_name –p –s 1 –M –C" options indicated. Gene stable IDs (GRCm38.p6) were downloaded from Ensembl Genes 94 (Ensembl) and used to filter for mouse protein coding genes. Matrix of raw counts from both genotypes across time was generated in R. Transcripts with less than eight counts across all samples were excluded from downstream analyses. All transcript counts were normalized using edgeR (*Robinson et al., 2010*) and expressed in counts per million (cpm).

Circadian expression of mRNA was determined using MetaCycle (*Wu et al., 2016*) with the following options: minper = 24, maxper = 24, cycMethod = c ("ARS," "JTK," "LS"), analysisStrategy = "auto," outputFile = TRUE, outIntegration = "both," adjustPhase = "predictedPer," combinePvalue = "fisher," weightedPerPha = TRUE, ARSmle = "auto," and ARSdefaultPer = 24. Transcripts with an integrated p-value (meta2d_pvalue) < 0.05 were considered rhythmically expressed.

Circadian pathways were determined by PSEA (*Zhang et al., 2016*) based on the sets of circadian transcripts with a relative amplitude value (rAMP; the ratio between amplitude and baseline expression) of ≥0.1. Gene sets were downloaded from the Molecular Signatures database (MSigDB) H (HallmarkGene Sets; *Subramanian et al., 2005*). Sets containing fewer than five circadian transcripts were excluded from the analysis. The Kuiper test was used to identify circadian gene sets by comparing the acrophases of all circadian transcripts belonging to each gene set to a uniform background distribution and by testing for non-uniformity (q < 0.01).

For the in vitro sugar challenge experiment, sequences were preprocessed using FASTQ Groomer and checked using FastQC (https://www.bioinformatics.babraham.ac.uk/projects/fastqc/). Reads were mapped against the *M. musculus* genome (mm10) downloaded from Ensembl.org using HISAT2with default parameters (*Kim et al., 2015*). After alignment to the genome, reads were counted with featureCounts (*Liao et al., 2014*) with default parameters. Matrices containing the counts were loaded into R. For downstream analyses, we analyzed protein coding genes with a sum >20 counts across all samples. Differential expression analysis between conditions was performed using the TestDE function in the R package 'edgeR' (*Robinson et al., 2010*). Genes with an FDR value <0.05 when comparing LG vs. HGF conditions in each respective genotype were considered differentially expressed and were

subsequently used to assess the pathways enriched under the LG and HGF conditions. Gene lists were uploaded to enrichR web server (*Chen et al., 2013*; *Kuleshov et al., 2016*), and pathway enrichment was assessed by comparing the overlap of genes in the query to the Kyoto Encyclopedia of Genes and Genomes (KEGG) 2019 mouse gene sets (*Kanehisa and Goto, 2000*).

For ChIP-seq read mapping, peak calling, and quantification, sequences were first preprocessed using FASTQ Groomer and checked using FastQC (https://www.bioinformatics.babraham.ac.uk/projects/fastqc/). PE reads were trimmed to 50 bases with Trimmomatic (*Bolger et al., 2014*), and reads were aligned to mouse genome (GRCm38/mm10) using Bowtie2 (*Langmead and Salzberg, 2012*) with the default parameters. Next, peak calls were made in MACS2 (*Feng et al., 2012*) with default parameters, except allowing for duplicates. Filters were applied to keep peaks with a log p-value <$-2$ and an Mfold >2. Results from the independent ChIP-seq runs were concatenated and intervals were merged using the MergeBED function from BEDtools (*Quinlan and Hall, 2010*). Identified peaks within the $-10$ kb to $+1$ kb region relative to the TSS were annotated using GREAT (*McLean et al., 2010*) and further filtered for the presence of a KLF10 motif (matrix ID: MA1511.1, http://jaspar.genereg.net/) using FIMO (*Bailey et al., 2015*).

All numerical values are presented as mean ± SEM. Replicates (n) are indicated in the figure legends. Pairwise comparisons were tested using the nonparametric Wilcoxon test. Differences between more than two experimental conditions were tested using the Kruskal–Wallis rank sum test for multiple comparison followed by a pairwise *post-hoc* test using the Benjamini–Hochberg adjustment. In *Figure 4I*, a three-way ANOVA was performed. p-values less than 0.05 were considered significant.

## Acknowledgements

This study was supported by French Agence Nationale pour la Recherche (ANR): #ANR-12-BSV1-0014 and by the "Investments for the Future" LABEX SIGNALIFE (#ANR-11-LABX-0028-01), the UCA[JEDI] Investments in the Future project (#ANR-15-IDEX-01) and ATER grant from Université Côte d'Azur (JSR). We thank the Canceropôle Provence-Alpes-Côte d'Azur and the Provence-Alpes-Côte d'Azur Region for the financial support provided to the MetaboCell project. We thank Pierre Chambon for the gift of *Alb-CreER*[T2] mice. We thank Aurélie Biancardini, Angelo Barnabeo, Pierre Chérel, Antoine Landouar, and the staff of the mouse facility for assistance with the animal care. We thank Sama Rekami for the help with the histopathology and the PFTC facility for technical support.

## Additional information

### Funding

| Funder | Grant reference number | Author |
|---|---|---|
| Agence Nationale de la Recherche | ANR-12-BSV1-0014 | Franck Delaunay<br>Michèle Teboul |
| Agence Nationale de la Recherche | ANR-11-LABX-0028-01 | Anthony A Ruberto<br>Franck Delaunay<br>Michèle Teboul |
| Agence Nationale de la Recherche | ANR-15-IDEX-01 | Franck Delaunay<br>Michèle Teboul |
| Canceropole PACA | MetaboCell | Mohamed Mehiri |
| Université Côte d'Azur | ATER | Johana S Revel |

The funders had no role in study design, data collection and interpretation, or the decision to submit the work for publication.

### Author contributions

Anthony A Ruberto, Investigation, Methodology, Writing - original draft, Writing - review and editing; Aline Gréchez-Cassiau, Johana S Revel, Mohamed Mehiri, Investigation, Methodology; Sophie Guérin, Luc Martin, Investigation, Methodology, Resources; Malayannan Subramaniam, Funding acquisition, Methodology, Resources, Writing - review and editing; Franck Delaunay, Conceptualization, Funding

acquisition, Investigation, Methodology, Project administration, Resources, Supervision, Writing - original draft, Writing - review and editing; Michèle Teboul, Conceptualization, Funding acquisition, Investigation, Methodology, Resources, Supervision, Writing - original draft, Writing - review and editing

### Author ORCIDs
Luc Martin ⓘ http://orcid.org/0000-0001-5725-3955
Franck Delaunay ⓘ http://orcid.org/0000-0003-4927-1701
Michèle Teboul ⓘ http://orcid.org/0000-0002-3418-4384

### Ethics
All animal studies were approved by the local committee for animal ethics Comité Institutionnel d'Éthique Pour l'Animal de Laboratoire (CIEPAL-Azur; Authorized protocols: PEA 244 and 557) and conducted in accordance with the CNRS and INSERM institutional guidelines.

### Decision letter and Author response
Decision letter https://doi.org/10.7554/eLife.65574.sa1
Author response https://doi.org/10.7554/eLife.65574.sa2

## Additional files

### Supplementary files
• Supplementary file 1. Primers used in the study .
• Transparent reporting form

### Data availability
Sequencing data have been deposited in European Nucleotide Archive under accession codes PRJEB39035, PRJEB39036, PRJEB40195. All data generated or analysed during this study are included in the manuscript and in source data files provided for Figures 1-7.

The following dataset was generated:

| Author(s) | Year | Dataset title | Dataset URL | Database and Identifier |
|---|---|---|---|---|
| Ruberto AA | 2021 | Genome-wide profiling of KLF10 binding loci in mouse liver | http://www.ebi.ac.uk/ena/data/view/PRJEB40195 | EBI, PRJEB40195 |
| Ruberto AA | 2021 | Transcriptional response of hepatocytes lacking KLF10 to a high sugar challenge | http://www.ebi.ac.uk/ena/data/view/PRJEB39036 | EBI, PRJEB39036 |
| Ruberto AA | 2021 | Circadian profiling of livers from control and hepatocyte-specific KLF10 knockout mice using RNA seq | http://www.ebi.ac.uk/ena/data/view/PRJEB39035 | EBI, PRJEB39035 |

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
