## [Decision Letter]

**Acceptance summary:**

This paper shows that loss of the KLF10 in hepatocytes results in reprogramming of the circadian transcriptome in liver and alters the temporal coordination of pathways associated with energy metabolism.

**Decision letter after peer review:**

Thank you for submitting your article "KLF10 integrates circadian timing and sugar signaling to coordinate hepatic metabolism" for consideration by *eLife*. Your article has been reviewed by 3 peer reviewers, one of whom is a member of our Board of Reviewing Editors, and the evaluation has been overseen by David James as the Senior Editor. The following individual involved in review of your submission has agreed to reveal their identity: Katja A Lamia (Reviewer #3).

Essential Revisions:

1. Further characterization of the hepatic Klf10KO model is needed to substantiate claims regarding altered circadian control of glucose and lipid metabolism. For example, it has been shown that animals deficient in transcription factors that regulate circadian metabolic genes demonstrate poor adaptation to prolonged fasting (e.g. Klf15, Srebp, etc). What are the effects of overnight fasting in these animals (this would further support the claims in Figure 2 and S2)? Are there changes in plasma chemistry throughout the dark-light cycle in the fed condition? Are there shifts in energy utilization throughout the dark-light cycle?

2. Figures 1 and 2 emphasize KLF10's role in regulation of the circadian transcriptome, but no further characterization of circadian regulation is pursued thereafter. While the title indicates that "circadian timing and sugar signaling" are "integrated", the second half of the paper fails to demonstrate any role of circadian timing in transcriptional regulation of metabolism. The authors should demonstrate how transcriptional changes in response to fructose/high glucose are altered with the circadian cycle.

3. Given the strong circadian expression of Klf10 and the title claim that KLF10 integrates circadian timing and sugar signaling, some of the analyses in Figures3-6 should be performed at multiple time points. Importantly, how does SSW impact circadian rhythms in the liver? Does it alter the daily profile of Klf10 mRNA and/or protein expression? Recruitment of KLF10 to chromatin at key loci? In figure 4, are all of these measurements performed at ZT15? The increase in G6pase upon exposure to SSW is surprising. I would expect that exposure to high sugar would decrease gluconeogenesis. Given the strong daily rhythms in expression of many of these genes, they should be measured at additional time points. At a minimum, expression should also be measured at ZT6-9, when KLF10 protein peaks under normal conditions (which is also the time of peak expression for Pck1 and G6pase).

4. How does the concentration of fructose used to treat hepatocytes (5 mM) relate to the amount of fructose present in tissues of mice or people after consumption of high fructose containing food/drinks? Please clarify in the text and methods section that SSW is the "sugar beverage" described in the methods. Given that the SSW treatment includes not only glucose + fructose but also sucrose, authors should check how sucrose influences expression of Klf10 in hepatocytes. As noted above in point 2, it would also be nice to see how the SSW consumption affects the concentrations of glucose, fructose, and sucrose to which hepatocytes are exposed in vivo. Related to this point, a highly relevant paper "Deletion of KLF10 Leads to Stress-Induced Liver Fibrosis upon High Sucrose Feeding" by J Lee (Int J Mol Sci 2020) should be cited and discussed.

5. The presentation and analysis of the data in Figure 3 are unclear. Given that the study involves two independent variables, it would be appropriate to use two-way ANOVA to analyze these data to evaluate whether there is a main effect of genotype, of SSW, and/or of an interaction between the two factors. Then, individual pairwise comparisons would be performed when a main effect is observed. For example, panel 3F is described as showing that "only KLf10Δhep mice displayed an increase in liver mass" but it is unclear from the data whether that conclusion is justified. It appears that SSW also causes increased liver mass in the control mice even if it does not reach statistical significance in the test used. That does not necessarily mean that the genotype has a significant impact on the effect of SSW on liver mass. Similarly, Figure 3H is described as showing that KLf10Δhep mice on SSW have a significant increase in glycemia but this seems to come from comparing the control and SSW conditions for Klf10∆*hep* mice while those mice seem to have lower glucose compared to the control mice in the absence of SSW treatment and it is unclear whether the overall statement is justified by the data. In Figure 3C, the horizontal lines showing which groups are being compared have almost no space between them so it is hard to see what is being compared. In Figure 3J, a repeated measures ANOVA should be used to compare the groups. In Figure 3K, the presentation is unclear. Do the symbols on the right apply to all pairwise comparisons that are "boxed"? So the p value is P<0.05 for both genotypes comparing control to SSW for hexose-6-phosphate for example?

6. The RNA-seq analysis in Figure 5C and 5D is confusing. It is not clear looking at the figure and the corresponding Supplemental figure which comparisons are being made. Is the comparison between genotypes for each treatment or between treatments for each genotype? Are the DEGs in the LG group achieved through comparing to an untreated group or to HGF (in which case, DEGs should only be represented once in relation to one treatment group)?

7. Figure 6D: The ChIP-seq data reveals widespread role for Klf10 in different metabolic networks. However, it is not clear why the authors compared the KLF10 ChIP-seq data with the RNA-seq DEGs comparing Klf10flox/flox and Klf10∆*hep* under HGF condition. If the authors want to know which genes are direct targets of Klf10, it would be more informative to compare the ChIP-seq data with the RNA-seq data set (at the same time point) from Figure 2 and then perform pathway analysis (this may yield hits for carbohydrate metabolism related pathways, which are not currently seen in 6D). In Figure 6B and 6D, are there really 7913 + 5317 = 13,230 KLF10 binding sites within the -10 to +1 kb region of only 213 + 93 = 306 genes? This seems hard to believe. Please clarify the discrepancy between these figures.

8. Previously published papers, including those from this group, have shown sex differences in the metabolic profile of Klf10-/- mice, thus exploring the phenotype resultant of specifically knocking out hepatocyte Klf10 in both male and female animals is of importance.

---

## [Author Response]

1. Further characterization of the hepatic Klf10KO model is needed to substantiate claims regarding altered circadian control of glucose and lipid metabolism. For example, it has been shown that animals deficient in transcription factors that regulate circadian metabolic genes demonstrate poor adaptation to prolonged fasting (e.g. Klf15, Srebp, etc). What are the effects of overnight fasting in these animals (this would further support the claims in Figure 2 and S2)? Are there changes in plasma chemistry throughout the dark-light cycle in the fed condition? Are there shifts in energy utilization throughout the dark-light cycle?

We have shown earlier that the response of systemic *Klf10* KO mice to a prolonged fasting is indistinguishable from that of WT mice in contrast to *Klf15* KO mice (Guillaumond et al., Mol Cell Biol 2010, PMID: 20385766; Gray et al., Cell Metab 2007, PMID**:** 17403374). We have also tested the response of hepatic *Klf10* KO mice to a 19 h overnight fast and found no significant difference between genotypes (112 ± 7 mg/dl in *Klf10^flox/flox^ vs* 113 ± 6 mg/dl in *Klf10^Δhep^* mice). This data is now reported in the result section. This is consistent with the observation that *Klf10* KO hepatocytes have an increased gluconeogenic capacity (see new supplemental Figure S2F). In Figure S2C and S2D we also show subtle changes in plasma glucose and hepatic glycogen levels during the resting phase but not during the active phase. Our view is that KLF10 and KLF15 seem to have different and complementary roles in the regulation of glucose homeostasis by regulating the hepatic response to fasting and feeding respectively. We unfortunately did not have the possibility to perform indirect calorimetry to determine if energy utilization is changed in *Klf10^Δhep^* mice during the LD cycle.

2. Figures 1 and 2 emphasize KLF10's role in regulation of the circadian transcriptome, but no further characterization of circadian regulation is pursued thereafter. While the title indicates that "circadian timing and sugar signaling" are "integrated", the second half of the paper fails to demonstrate any role of circadian timing in transcriptional regulation of metabolism. The authors should demonstrate how transcriptional changes in response to fructose/high glucose are altered with the circadian cycle.

We agree that this is an important point which is now addressed in Figure 4 and Figure 5 where we have included gene expression data at both ZT9 and ZT15 peak and trough of *Klf10*, respectively. In Figure 4, we show that *Klf10* is inducible by sugars at ZT15 but not at ZT9. Figure 5 shows that in *Klf10^Δhep^* mice the chow + SSW diet changes the expression of lipogenic genes at ZT15 but not ZT9 while genes regulating glycolysis and gluconeogenesis are impacted at both ZTs.

3. Given the strong circadian expression of Klf10 and the title claim that KLF10 integrates circadian timing and sugar signaling, some of the analyses in Figures3-6 should be performed at multiple time points. Importantly, how does SSW impact circadian rhythms in the liver? Does it alter the daily profile of Klf10 mRNA and/or protein expression? Recruitment of KLF10 to chromatin at key loci? In figure 4, are all of these measurements performed at ZT15? The increase in G6pase upon exposure to SSW is surprising. I would expect that exposure to high sugar would decrease gluconeogenesis. Given the strong daily rhythms in expression of many of these genes, they should be measured at additional time points. At a minimum, expression should also be measured at ZT6-9, when KLF10 protein peaks under normal conditions (which is also the time of peak expression for Pck1 and G6pase).

As indicated for point 2 above, we are now providing data showing the impact of the chow + SSW diet on the expression of *Klf10* determined at ZT9 and ZT 15 corresponding to its peak and through (see Figure 4B in the revised manuscript). In Figure 5 we have also analyzed the expression of *Slc2a4, Pklr, Pck1, Fasn* and *Elovl6* at these time points. Interestingly, this data now shows that the impact of the hepatic *Klf10* knockout combined with the high sugar diet changes the expression of lipogenic genes in a time-dependent manner while genes regulating glycolysis and gluconeogenesis are impacted at both ZTs.

The experiment reported in Figure 5 (Figure 6 in the revised version) was performed with primary hepatocytes and not designed to allow a RNA-seq profiling around the clock.

The recruitment of KLF10 to chromatin loci was investigated in unchallenged mice, at ZT9 in as reported in Fig7 A-D. When upregulated upon feeding with chow+ SSW, KLF10 also binds to *Acss2* and *Acacb* at ZT15 (Figure 7E).

We agree that induction of the gluconeogenic gene *G6Pase* in mice fed chow + SSW seems paradoxical at first glance. However, this effect has been consistently observed in earlier studies (see for instance Argaud et al., J Biol Chem, 1997, PMID 9139747; Sun et al., J Biol Chem, 2019, PMID: 31481463; Gautier-Stein et al., Diabetes, 2012, PMID: 22787137). Notably, G6Pase is also augmented in diabetic mice and in type 2 diabetes patients (Clore JN et al., Diabetes 2000 PMID: 10866049 ; Gautier-Stein et al., Diabetes, 2012, PMID: 22787137). The current explanation for this effect is that glucotoxicity resulting from high sugar consumption or insulin resistance leads to an increased ROS production which in turn stimulates the transcriptional activity of the transcription factor HIF1a which is a positive regulator of the G6Pase promoter (Gautier-Stein et al., Diabetes, 2012, PMID: 22787137).

4. How does the concentration of fructose used to treat hepatocytes (5 mM) relate to the amount of fructose present in tissues of mice or people after consumption of high fructose containing food/drinks? Please clarify in the text and methods section that SSW is the "sugar beverage" described in the methods. Given that the SSW treatment includes not only glucose + fructose but also sucrose, authors should check how sucrose influences expression of Klf10 in hepatocytes. As noted above in point 2, it would also be nice to see how the SSW consumption affects the concentrations of glucose, fructose, and sucrose to which hepatocytes are exposed in vivo. Related to this point, a highly relevant paper "Deletion of KLF10 Leads to Stress-Induced Liver Fibrosis upon High Sucrose Feeding" by J Lee (Int J Mol Sci 2020) should be cited and discussed.

We have clarified the text and method section regarding the definition of the SSW beverage. At the organism level, the small intestine absorbs only monosaccharides. Sucrose being a disaccharide, it is first cleaved into glucose and fructose by the digestive enzyme sucrase. Unless we misunderstood this comment, looking at the effect of sucrose by itself in isolated hepatocytes on the expression of *Klf10* is therefore irrelevant*.* Measuring exactly how SSW consumption impacts on the concentration of fructose and glucose circulating in the liver would require metabolic flux analysis which was beyond the scope of our study. Further, as said above, sucrose is converted into glucose and fructose. The Rabinowitz’s lab has shown that dietary fructose is cleared by the small intestine upon conversion into glucose and glycerate (and other metabolites at lower concentration). Fructose derived glucose and glycerate are then found in the hepatic portal vein. However, upon excess of dietary fructose, the small intestine is overwhelmed and fructose is transported to hepatocytes by the portal blood (Jang et al., Cell Met 2018, PMID: 29414685). We agree that the paper by Lee et al., is highly relevant to our study and have cited and discussed it in the revised manuscript.

5. The presentation and analysis of the data in Figure 3 are unclear. Given that the study involves two independent variables, it would be appropriate to use two-way ANOVA to analyze these data to evaluate whether there is a main effect of genotype, of SSW, and/or of an interaction between the two factors. Then, individual pairwise comparisons would be performed when a main effect is observed. For example, panel 3F is described as showing that "only KLf10Δhep mice displayed an increase in liver mass" but it is unclear from the data whether that conclusion is justified. It appears that SSW also causes increased liver mass in the control mice even if it does not reach statistical significance in the test used. That does not necessarily mean that the genotype has a significant impact on the effect of SSW on liver mass. Similarly, Figure 3H is described as showing that KLf10Δhep mice on SSW have a significant increase in glycemia but this seems to come from comparing the control and SSW conditions for Klf10∆hep mice while those mice seem to have lower glucose compared to the control mice in the absence of SSW treatment and it is unclear whether the overall statement is justified by the data. In Figure 3C, the horizontal lines showing which groups are being compared have almost no space between them so it is hard to see what is being compared. In Figure 3J, a repeated measures ANOVA should be used to compare the groups. In Figure 3K, the presentation is unclear. Do the symbols on the right apply to all pairwise comparisons that are "boxed"? So the p value is P<0.05 for both genotypes comparing control to SSW for hexose-6-phosphate for example?

Figures 3, 4 and 5 (now Figures 3, 4, 5 and 6 in the revised manuscript) report experiments investigating the effect of two independent variables (genotype and diet). In all these experiments the number of replicates is too small (n=3-12) to guarantee that we do have a normal distribution of the data, which is the prerequisite for using parametric statistical tests including the two-way ANOVA. For this reason, as stated in the quantification and statistical analyses section, we have used the Kruskal-Wallis non parametric test followed by pairwise post-hoc testing using the Benjamini-Hochberg adjustment, which is the equivalent of a one-way ANOVA. We agree that a limitation of non-parametric tests is that an interaction between the two variables cannot be detected. However, when a statistical difference is detected between the two genotypes on the chow + SSW diet but not on the chow diet, we are quite confident that this difference is the result of a combined effect of the mutation and the diet. We agree that regarding Figure 3F (Figure 4E in the revised manuscript), the statement “only KLf10^Δhep^ mice displayed an increase in liver mass” could confuse the readers. We have therefore replaced this sentence by “We also observed a trend for an increase in liver mass of mice fed a chow +SSW diet that reached significance in Klf10^Δhep^ mice (Figure 4E)”.

Figure 3C (Figure 4B in the revised manuscript). The layout has been improved.

Figure 3H (Figure 4G in the revised manuscript). The point for this figure is that *Klf10^Δhep^* mice on chow have a slightly but not significantly (*p* = 0.225) lower glycemia than the *Klf10^flox/flox^*. In addition, no difference was detected between the two diets for *Klf10^flox/flox^* mice (p = 0.624). Conversely, we do detect a statistically higher glycemia in *Klf10^Δhep^* mice on chow + SSW compared to their controls on chow (*p* = 0.0353). To more accurately describe this data we have replaced the sentence in the result section “Compared to *Klf10^flox/flox^* mice, *Klf10^Δhep^* mice on the chow + SSW diet had a significant increase in glycemia” by “Glycemia was moderately increased in *Klf10^Δhep^* mice on chow diet as compared to their control on chow”.

Figure 3J (Figure 4I in the revised manuscript). In the GTT experiment for which we have a large dataset (Fig5I), we performed a three-way ANOVA and we could detect a significant interaction between the genotype and diet as reported in the result section.

Figure 3K (Figure 4J in the revised manuscript). The symbols next to the metabolite names apply to all the comparisons indicated by the boxes in the heatmap. We have clarified this in the figure legend.

6. The RNA-seq analysis in Figure 5C and 5D is confusing. It is not clear looking at the figure and the corresponding Supplemental figure which comparisons are being made. Is the comparison between genotypes for each treatment or between treatments for each genotype? Are the DEGs in the LG group achieved through comparing to an untreated group or to HGF (in which case, DEGs should only be represented once in relation to one treatment group)?

The experiment reported in Figure 5 (Figure 6 in the revised manuscript) aimed at determining whether the transcriptional response of primary hepatocytes exposed to a shift from a low glucose (LG) containing medium to a high glucose/fructose (HGF) containing medium is changed in the absence of KLF10. To clarify which comparisons were made, we have improved the readability of the volcano plots in Figure 6C so that the reader can better understand that this panel displays DEGs in the HGF vs LG treatment for each genotype. As now indicated on the right of Figure 6C, downregulated genes in HGF are upregulated in LG (and *vice versa*). In Figure 6D, we use this DEG output (see table S3 for gene lists) in a pathway enrichment analysis to determine which metabolic processes are activated in *Klf10^flox/flox^* and *Klf10^Δhep^* hepatocytes when treated with either LG or HGF. The result of this analysis is presented based on significantly enriched pathways in the LG condition (down regulated genes in HGF) and in the HGF condition (upregulated genes in HGF) separately so that differences in the metabolic response between genotypes can easily be visualized.

7. Figure 6D: The ChIP-seq data reveals widespread role for Klf10 in different metabolic networks. However, it is not clear why the authors compared the KLF10 ChIP-seq data with the RNA-seq DEGs comparing Klf10flox/flox and Klf10∆hep under HGF condition. If the authors want to know which genes are direct targets of Klf10, it would be more informative to compare the ChIP-seq data with the RNA-seq data set (at the same time point) from Figure 2 and then perform pathway analysis (this may yield hits for carbohydrate metabolism related pathways, which are not currently seen in 6D). In Figure 6B and 6D, are there really 7913 + 5317 = 13,230 KLF10 binding sites within the -10 to +1 kb region of only 213 + 93 = 306 genes? This seems hard to believe. Please clarify the discrepancy between these figures.

The experiments reported in Figure 3-5 support a role for KLF10 in energy metabolism. Further, the experiment performed with primary hepatocytes revealed that KLF10 extensively modulates the transcriptional response to hexose sugars. To keep the focus of the paper on energy metabolism, we integrated the data obtained with the RNA-seq experiment comparing the response of *Klf10^flox/flox^* and *Klf10^Δhep^* primary hepatocytes to a high sugar treatment with the data obtained with the ChIP-seq experiment performed in *Klf10^flox/flox^* liver. One advantage of using the primary hepatocyte RNA-seq data is that we are looking at a cell autonomous transcriptome response to the metabolic challenge. Additionally, the transcriptome that we determined in Figure 2 may be not so informative as the animals were unchallenged while the metabolic phenotype is mostly visible in high sugar fed animals as reported in the paper. Further, the difference we see between the circadian transcriptomes of *Klf10^flox/flox^* and *Klf10^Δhep^* mice as revealed by the PSEA, mainly reflects a compromised temporal coordination, therefore using a single time point for constructing the network may lead to a bias in the analysis. Finally, the time course RNA-seq in liver was done using pooled samples (3 per time point) which precludes statistical analysis when using one single time point whereas the triplicates available with the experiment in challenged hepatocytes allowed such statistical analysis and subsequent integration of the data.

The bargraph shown in Figure 6B (Figure 7B in the revised manuscript) reports the number of KLF10 binding sites identified in the -50/+1 kb region of putative target genes. The 13,230 KLF10 binding sites identified in the -10/+1 kb proximal region are present in 7,898 unique genes. The full HumanCyc network that we used in our analysis contains 6,161 nodes including 2,566 genes and 81,590 edges. As the list of genes bound by KLF10 only partially intersect with the gene list of this metabolic network, the 306 KLF10 bound genes reported in Figure 6D (Figure 7D in the revised manuscript) and supplemental table S5 should not be directly compared to the number of binding sites reported in Figure 6B but instead considered as the fraction (12 %) of the metabolic genes of the HumanCyc network that are potentially regulated by KLF10. To better visualize this, we have modified the scheme in Figure 7A to indicate that the metabolic network in Figure 7C results from the merging of ChIP-seq and DEGs data with the HumaCyc metabolic network. We have also clarified the associated text in the results and Discussion sections.

8. Previously published papers, including those from this group, have shown sex differences in the metabolic profile of Klf10-/- mice, thus exploring the phenotype resultant of specifically knocking out hepatocyte Klf10 in both male and female animals is of importance.

We agree this is definitely an important point. Accordingly, during the initial step of the project we collected liver samples from female mice around the clock to perform some pilot experiments comparing the impact of the hepatocyte specific loss of KLF10 on circadian gene expression between males and females. We measured expression of the *Elovl6* mRNA as a marker of fatty acid synthesis and for which the systemic deletion of *Klf10* caused an opposite phase shift between males and females resulting in an approximately 7 h phase difference (Guillaumond et al., Mol Cell Biol, 2010, PMID: 20385766). As seen in Author response image 1, we did not observe this dramatic difference in the phase of *Elovl6* oscillation between male and female *Klf10^Δhep^* mice in which the acrophase of *Elovl6* was similar. Based on this result, we hypothesize that the sex difference seen in the systemic *Klf10* knockout model may be attributable to the lack of KLF10 in extrahepatic tissues or non-parenchymal hepatic cells. Of course, we cannot exclude that some sex difference exists in *Klf10^Δhep^* animals but not retrieving a major difference seen in the Klf10 KO model led us to not further explore the phenotype of *Klf10^Δhep^* female mice in this study.

**Author response image 1. sa2fig1:**